



# Refined estimates of water transport through the Åland Sea, Baltic Sea

Antti Westerlund[1], Elina Miettunen[2], Laura Tuomi[1], and Pekka Alenius[1]

[1]Finnish Meteorological Institute, Marine Research, Erik Palménin aukio 1, P.O.Box 503, FI-00101 Helsinki, Finland
[2]Finnish Environment Institute SYKE, Marine Research Centre, Latokartanonkaari 11, FI-00790 Helsinki, Finland
**Correspondence:** Antti Westerlund (antti.westerlund@fmi.fi)

**Abstract.** Water exchange through the Åland Sea, Baltic Sea, greatly affects the environmental conditions in the neighbouring Gulf of Bothnia. Recently observed changes in the eutrophication status of the Gulf of Bothnia may be connected to changing nutrient fluxes through the Åland Sea. Pathways and variability of sub-halocline northward-bound flows towards the Bothnian Sea are important for these studies. While the general nature of the water exchange is known, that knowledge is based on only a

few studies that are somewhat limited in details. Notably, no high-resolution modelling studies of water exchange in the Åland Sea area have been published. In this study, we present a configuration of the NEMO 3D hydrodynamic model for the Åland Sea–Archipelago Sea area at around 500 m horizontal resolution. We then use it to study the water exchange in the Åland Sea. We first ran the model for the years 2013–2017 and validated the results, with a focus on the simulated current fields. We found that the model reproduced current direction distributions and layered structure of currents in the water column with

reasonably good accuracy. Next, we used the model to calculate volume transports across several transects in the Åland Sea. These calculations provided new detail of water transport in the area. Time series of monthly mean volume transports showed a consistent northward transport in the deep layer. In the surface layer there was more variability: while net transport was towards the south, in several years some months in late summer or early autumn showed net transport to the north. Furthermore, based on our model calculations, it seems that dynamics in the Lågskär Deep are more complex than has been previously understood.

While Lågskär Deep is the primary route of deep water exchange, still a significant volume of deep water enters the Åland Sea through the depression west of the Lågskär Deep. Better spatial and temporal coverage of current measurements is needed to further refine the understanding of water exchange in the area.

## 1  Introduction

The Gulf of Bothnia, in the northern Baltic Sea, has so far been in relatively good environmental health and free from both

seasonal and long-term hypoxia occurring in many other Baltic Sea basins. Recently evaluated long-term trends, however, show changes in the eutrophication status (Kuosa et al., 2017). Reasons for these changes are currently not fully understood. One piece of the puzzle are the still poorly understood fluxes of nutrient rich and also possibly hypoxic water from the Baltic Proper in the south (Ahlgren et al., 2017).



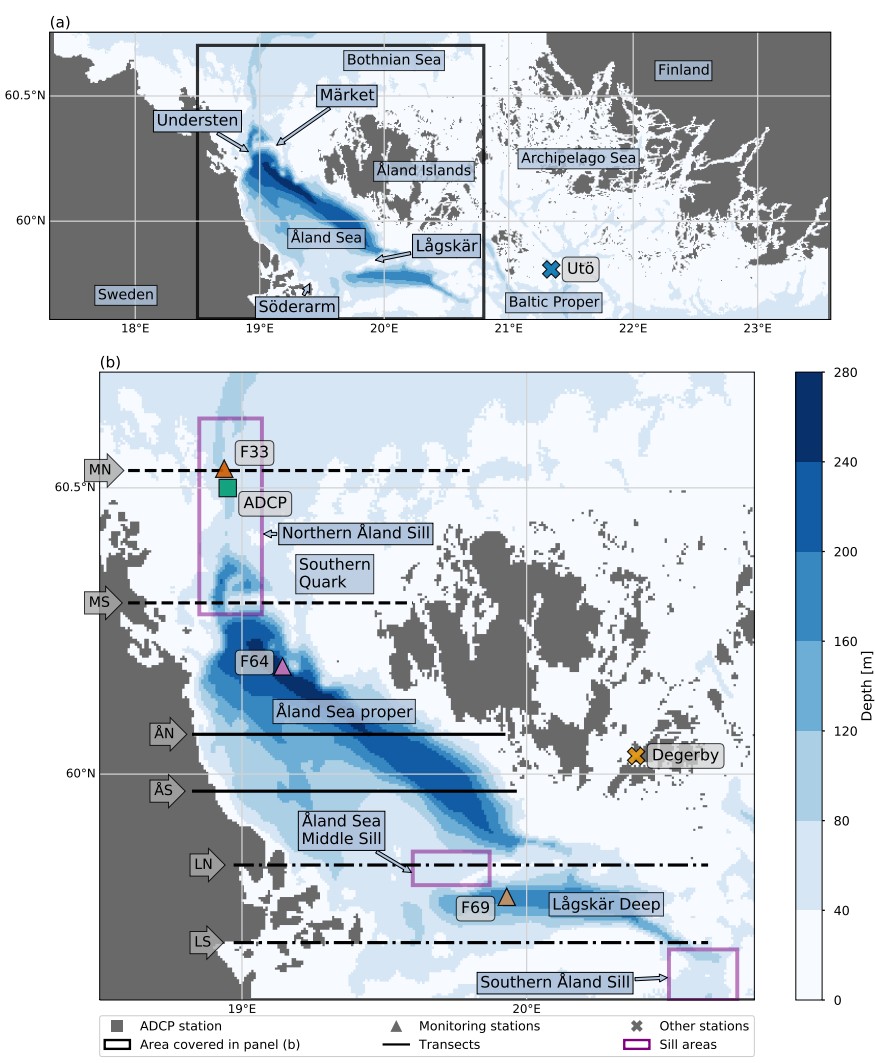

**Figure 1.** (a) Model domain and bathymetry. Main focus area of this study is shown with a rectangle. Also, a number of geographic references are displayed. (b) A more detailed map of the main focus area. Sub-basins and sill areas relevant for the analysis are shown. Furthermore, locations of stations and transects mentioned in the text are indicated.



Water has two main routes between the Baltic Sea Proper and the Gulf of Bothnia: the deeper, but narrower Åland Sea, and
the wider but shallower Archipelago Sea (Fig. 1). A series of sills and smaller sub-basins regulate the exchange in the Åland
Sea route, and numerous islands and narrow passages control the exchange in the Archipelago Sea route. Depths in the area
vary from just a few metres in the shallow archipelago area to three hundred metres in the Åland Sea, with many relatively
steep gradients along the bottom (Leppäranta and Myrberg, 2009). The mean depth of the Archipelago Sea is only 19 m. It
is characterized by numerous small islands and narrow straits. The Åland Sea has two basins. The smaller Lågskär Deep (or
Lågskär Basin, or the southern Åland Sea basin) has the maximum depth of 220 m, and the larger Åland Sea proper (or the
northern Åland Sea basin) has the maximum depth of 301 m. Three sills affect water exchange through the Åland Sea. The
southernmost of these is the Southern Åland Sill, which is a narrow channel on the southern edge of the Åland Sea. It is a
major barrier for deep water entering from Baltic Proper. Next is the Åland Sea Middle Sill between Söderarm and Lågskär,
separating the Lågskär Deep from Åland Sea proper. The third is the Northern Åland Sill in the Southern Quark area located
at the northern edge of the Åland Sea.

Complex bathymetric features, significant depth variations and patchy observational data pose challenges for studying the
exchanges between Baltic Proper and the Gulf of Bothnia. Likewise, previous modelling efforts have been hindered by insuf-
ficient local resolution and unresolved bathymetric features. In this paper, we present a new high-resolution three-dimensional
hydrodynamic modelling configuration for the Åland Sea and the Archipelago Sea. This configuration provides a platform for
studying dynamics in the area. We focus on the Åland Sea to investigate the exchanges through the area.

Probably the first comprehensive study of water exchange between the Baltic Proper and Gulf of Bothnia was by Witting in
1908. In early 1920's Åland Sea studies were continued as Finnish–Swedish co-operation with both hydrographic and current
meter observations in 1922 and 1923. A number of hydrographic surveys were conducted in the Åland Sea in the 20th century
(e.g Lisitzin, 1951). In late 1950's Hela (1958) estimated the water exchange between the Gulf of Bothnia and the Baltic
Proper using hydrographic data from the area measured in 1956. As also in the earlier Baltic Sea studies by Granqvist in 1938,
Hela (1958) found that there were no clearly defined water masses, and continuous mixing occurs between waters of different
origins. He also noted the existence of a deep water type, which originates from water entering from the Baltic Proper. The
term "deep water" was used to emphasize that this water was not Baltic Proper bottom water, as the most saline bottom water
is not able to flow over the Southern Åland Sill. Also notable were the warm water lenses below the thermocline and above the
temperature minimum, which Hela (1958) analyzed to be warm water intrusions from Baltic Proper. This analysis was later
further elaborated by Hela (1973).

Palosuo (1964) evaluated the water exchange using bathymetric information to study the channels and sill depths on the
deeper paths leading from the Baltic Proper to Gulf of Bothnia through Lågskär Deep and Åland Sea. He also concluded from
data from station F64 that at Lågskär Deep there is large variability in the salinity in the surface layer up to 80–90 m depth,
below which salinity increases constantly to values up to 8 g kg$^{-1}$. The Åland Sea proper showed similar salinity variation at
surface layer and more constant values in the deeper layers, being less saline than in the Lågskär Deep. A constant northward
current was estimated to be present in the deeper layer of the Åland Sea proper based on the datasets both in Palosuo (1964)
and Hela (1958).





After these studies the most recent in-depth investigations concentrating on the water exchange in the Åland Sea were
produced within the framework of the Finnish-Swedish co-operation to investigate the environmental condition of the Gulf of
Bothnia in the 1970's (Ehlin and Ambjörn, 1977; Ambjörn and Gidhagen, 1979).

These earlier studies relied heavily on observational and mostly short term datasets. More recently, numerical modelling has
been used to supplement our understanding. The promise of modelling for these kinds of studies is in its ability to describe
currents and water exchanges fully in four dimensions, giving us spatial and temporal coverage not possible with just observa-
tions. This allows us to investigate intra- and inter-annual variability, for example, with much richer detail than we could with
just observations.

Modelling studies investigating transports in this area have so far been rare. A notable exception is the study by Myrberg and
Andrejev (2006). Although they concentrated mainly on the Gulf of Bothnia as a whole, they also looked at water exchange
in the Åland Sea – Archipelago Sea area. They found that transport estimates had significant uncertainty and depended on the
choice of averaging time-scales for velocities and transports, as well as on the chosen locations of the cross-sections used for
inflow and outflow estimates.

Our modelling configuration is based on the NEMO model core (Madec and NEMO System Team, 2019), which has pre-
viously been applied in high-resolution in the nearby Gulf of Finland (Vankevich et al., 2016; Westerlund et al., 2018, 2019).
The Archipelago Sea has been previously modelled in high-resolution by Tuomi et al. (2018) and Miettunen et al. (2020)
who used the COHERENS model (Luyten, 2013). The configuration presented in this study builds on that experience, but
also covers the Åland Sea, which was not included in the COHERENS setup. In their study Tuomi et al. (2018) found that
the $\sigma$ coordinates used in the COHERENS implementation introduced over-mixing especially in deep channels where there
were big depth gradients from one grid point to the next. To mitigate this issue we chose the z* vertical coordinate system for
our implementation. This is a geopotential vertical coordinate system where sea surface height variations are distributed over
the whole water column (e.g. Klingbeil et al., 2018). It has been successfully applied in the aforementioned Gulf of Finland
configuration, as well as in regional configurations (Hordoir et al., 2019).

Bathymetric features greatly affect water exchange in this complicated, topographically irregular area. Forming a correct
understanding requires detailed information about pathways available for water and depth of sills. In principle bathymet-
ric data is available from different sources with high spatial resolution, see e.g. Baltic Sea Bathymetry Database (BSBD)
available at data.bshc.pro, or European Marine Observation and Data Network (EMODnet) bathymetry available at www.
emodnet-bathymetry.eu. But the resolution and accuracy of the source data is not equally good in all areas, see e.g. Jakobsson
et al. (2019) for an example from the Åland Sea. For this paper, we took significant effort to ensure that the bathymetry used
in the model represents bathymetric features in key areas as accurately as possible.

In this paper we first introduce the new modelling configuration. Then, a basic validation of a five year model run is carried
out. After that, we investigate modelled currents and finally study volume transports through the study area.





## 2 Materials and methods

### 2.1 Modelling methods

We set up the NEMO three-dimensional hydrodynamic model version 4.0.3 for the Åland Sea and the Archipelago Sea at 0.25 NM (nautical mile) or approximately 500 m horizontal resolution. Model domain and bathymetry is depicted in Fig. 1. The modelled time span covered June 2012 to December 2017, but in our analysis we concentrate on results starting from January 2013 to ensure that the model had a long enough initialization period.

The model setup used the z* vertical coordinate system. There were 200 vertical levels. Level thickness increased slightly with depth, being 1 m at the surface and 1.1 m at 120 m depth. Below the depth of 120 m the thickness increased more rapidly to about 8 m at the very bottom. This arrangement allowed for the top part of the water column to be resolved at a relatively high resolution, while keeping the number of levels manageable.

Horizontal viscosity was parameterized with the Smagorinsky formulation (Smagorinsky, 1963). Smagorinsky formulation has been a popular choice for studying nearby sea areas, see e.g. Zhurbas et al. (2008). We set the free Smagorinsky parameter as defined in NEMO (see Madec and NEMO System Team, 2019) to `rn_csmc=3.0`, which we found to produce reasonable results. As e.g. Holt and James (2006) note after conducting a model sensitivity experiment with the Smagorinsky formulation in British shelf sea waters, the choice of the free parameter is based more on expert judgement than on theoretical considerations or empirical evidence. In the vertical, we used the GLS (Generic Length Scale) mixing scheme configured to produce the k-$\epsilon$ model. This parameterization has previously been successfully applied in Baltic Sea NEMO configurations, and it has been able to reproduce seasonal stratification in the Bothnian Sea quite well (Westerlund and Tuomi, 2016).

We used an ice model with thermodynamic formulation (as was previously done for the Gulf of Finland by Westerlund et al., 2018, 2019). This somewhat eased the relatively high computational requirements of this configuration. During our study period, the ice seasons in the Baltic Sea area were mostly mild or very mild. In the Åland Sea, within our modelling period, there was a notable amount of ice only during winter 2012–2013, which has been classified as an average ice season by FMI (Finnish Meteorological Institute). 2017–2018 was also an average ice season, but the ice in the area formed only after our modelling period.

We saved 6 hour averages of 3D temperature, salinity and current fields. Sea surface height was recorded at one hour interval, while volume transports were saved once a day. We computed volume transports from the model for a number of transects. These were integrated over the whole transect to calculate a time series: $F_v = \iint v \, dA = \iint v \, dz dl$. Here $v$ is velocity across the transect, $A$ is area of the transect, $z$ is the depth along the transect and $l$ is the length of the transect. We also calculated volume transports per unit length ($\int v \, dz$) along the transects to investigate the pathways of water more closely.

### 2.2 Bathymetric data

We compiled the model bathymetry from two sources. The primary source for bathymetric data was the VELMU (Finnish Inventory Program for the Marine Environment) bathymetry model (Finnish Environment Institute), which covers the Finnish Exclusive Economic Zone (EEZ). For the part of model domain that is outsize the Finnish EEZ, we used the BSBD from Baltic



Sea Hydrographic Commission, which covers the whole modelling domain. The resolution of VELMU bathymetry model is approximately 20 m but the resolution of its source data is not as high in all locations. Same applies for any other gridded bathymetry dataset. The bathymetry data for the 0.25 NM model grid was compiled by calculating the mean of VELMU depth points in each model grid point. BSBD data has the resolution of 0.25 NM in the Åland Sea.

Because the source datasets have mostly been created and interpolated automatically and the compilation process of our model grid from those datasets was automatic, the resulting grid was somewhat patchy. For example, the channels crossing the area were not all continuous or deep enough, the sills were typically too shallow, and there were some very steep depth gradients that would be very challenging for the hydrodynamic model. To mitigate these issues we checked and edited the coastline, the locations of islands and the sill depths in the model grid manually to ensure that it represented the coastline and the depth variations in the area as accurately as possible in the 0.25 NM resolution.

The most crucial places to be modified were the sills and channels that control the water exchange through the Åland Sea: the Southern Åland Sill near the southern edge of the model domain and the channel leading to the Lågskär Deep from there; the Åland Middle Sill between Söderarm and Lågskär and the other pathways between the Åland Sea basins in the eastern side of Lågskär; and the sill between Understen and Märket and the Northern Åland Sill in the northern part of the Åland Sea (Fig. 1). Each sill area and its surroundings were checked manually and edited to equal the known sill depths. The channels in the southern and northern parts of the Åland Sea were also edited to be continuous at certain depth and overall wide enough (at least 3–4 grid points) so that they would enable appropriate water flow between the basins.

Finally, the modified model depth grid was filtered with Gaussian filter with standard deviation of 1.2 grid points to smooth out the steepest bathymetry gradients.

## 2.3 Forcing and boundary conditions

We subset the meteorological forcing from the ERA5 atmospheric reanalysis provided by Copernicus Climate Change Service (Hersbach et al., 2018). We used hourly 10-metre winds, 2-metre air temperature, 2-metre dewpoint temperature, mean sea level pressure, precipitation, snowfall rate, short wave radiation flux and long wave radiation flux fields from the reanalysis to force the model.

The model configuration had open boundaries to the Bothnian Sea in the north and to Baltic Proper and Gulf of Finland in the south. We took lateral boundary conditions and initial conditions for the model from a regional reanalysis configuration (Baltic Sea Physical Reanalysis Product `BALTICSEA_REANALYSIS_PHY_003_011`), provided by the Copernicus Marine Environment Monitoring Service (CMEMS). Initial conditions consisted of interpolated salinity and temperature fields. Boundary conditions included Flather radiation conditions for sea surface heights at one hour interval and barotropic velocities at 24 hour intervals. FRS (Flow Relaxation Scheme) boundary conditions were applied for temperature and salinity at one day interval. For the small open sea segment in the south-east edge of the model domain a no-flux condition was applied, which improved stability of the configuration. This area is quite shallow and far away from the area of interest in this study, so a no-flux condition was deemed sufficient for this purpose.





There are eight rivers inside the model domain, all on the Finnish coast. We took daily values of river discharge from the watershed model VEMALA which is an operational, national-scale nutrient loading model for Finnish watersheds (Huttunen et al., 2016).

### 2.4 Observational data

We used sea surface height from the Föglö Degerby station (location see Fig. 1). There are also two other tide gauges within the model domain in Turku and Forsmark. However, we did not use data from these sites as they are not representative of the overall sea level variation in the study area, but rather reflect local effects.

We also investigated temperature and salinity profiles from three stations in the Åland Sea: F33, F64 and F69 (locations see Fig. 1). These sites are sampled more often and their temporal coverage is better than in other stations in the area. However, the number of profiles is still quite modest and we supplemented them with data from the Utö intensive monitoring station at the southern edge of the Archipelago Sea.

We used current measurement data from a location near the station F33 to study how the model is able to reproduce observed currents (station "ADCP" in Fig. 1). The measurements were carried out with a bottom-mounted 300 kHz Workhorse Sentinel acoustic Doppler current profiler (ADCP). The location was 126 m deep and the corresponding model grid point was 110 m deep. The measurements ranged vertically from 8 m depth down to 112 m depth at 2 m intervals. This dataset covers the time period of 6 August 2016–3 July 2018 with a time interval of 30 min.

The modelled current components were saved as means of 6 hours. For the comparison between the measured and modelled currents, we first calculated 6 hour means of the measured horizontal current components and then from these means, calculated the current magnitude and direction. The ADCP data has been quality checked, leaving gaps in the dataset. These gaps due to bad or missing data occurred mainly in the upper 40 m layer during spring and summer. If a 6 h time slot over which the means were calculated was missing more than 50 % of the measurements, we discarded the time slot both from the averaged ADCP data and from the model data when calculating the bias and current roses.

## 3 Results

### 3.1 Model validation

#### 3.1.1 SSH

Adequate accuracy of sea surface heights is required for modelling several other processes. Therefore, comparison of modelled sea surface height with tide gauge data gives a quick overview of overall model performance. It is good to bear in mind, though, that these results are heavily dominated by how well SSH is presented in the boundary conditions.

The results from the Föglö Degerby station (location see Fig. 1) showed that the model was able to reproduce sea level variations quite well. A part of the SSH time series is displayed in Fig. 2 showing typical results. It shows that timing of sea

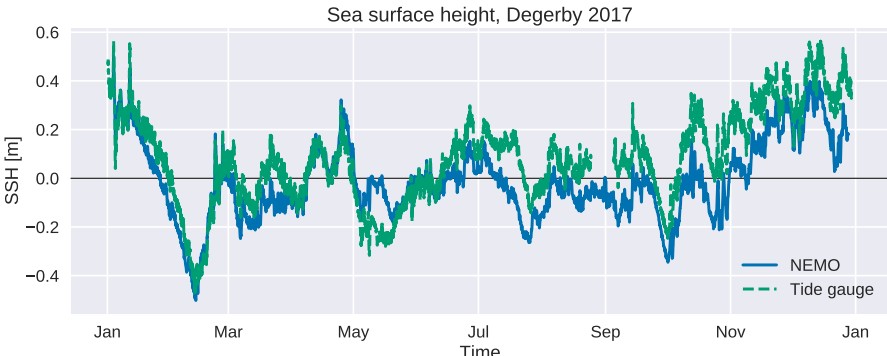

**Figure 2.** Sea surface height at the Föglö Degerby tide gauge in 2017, comparison between measurement and model.

level events is quite accurate. There are a few cases where the magnitude of the event was incorrectly estimated in the model, although in most cases the differences are quite small and the range of water level variability is overall well represented.

### 3.1.2 Temperature and salinity

Vertical structure of the water column is illustrated with observed and modelled temperature and salinity profiles from three monitoring stations in Fig. 3, which shows all available profiles from these stations along with overall mean profiles. Additionally, we compared profiles individually to model data to get an overview of model performance. Vertical temperature profiles are typically quite well reproduced. This includes the seasonal thermocline and its location, which are relatively correctly estimated. Salinity profiles reveal that salinity biases are most prevalent near the surface, with typical differences up to 1 g kg$^{-1}$

above the halocline. In some cases, there is evidence in the profiles of submesoscale activity, which is very difficult to fully capture due to the chaotic nature of such phenomena. Overall, the moderate availability of profile data from the modelling area somewhat limits the conclusions that can be drawn based on this data. Despite of this, the ability of the model to reproduce vertical temperature and salinity structure seems to be well within expected accuracy for a state-of-the-art model and the area of interest.

In addition to this, we inspected temperature and salinity time series from the Utö intensive monitoring station at the southern edge of the Archipelago Sea (not shown). Overall, these results indicate very similar model skill as has previously been reported for the NEMO model in nearby sea areas (see e.g. Westerlund and Tuomi, 2016; Westerlund et al., 2018). Temperature evolution is quite well reproduced at all depths and seasons, although there are larger differences deeper in the water column. While the frequency of the observations does not allow for a detailed analysis of short-term variability, it does seem that at least some

shorter-term events are also reproduced by the model. Salinity observations do not show the same kind of short-term variability as the modelled time series. The general level of modelled salinity values is quite sensible.


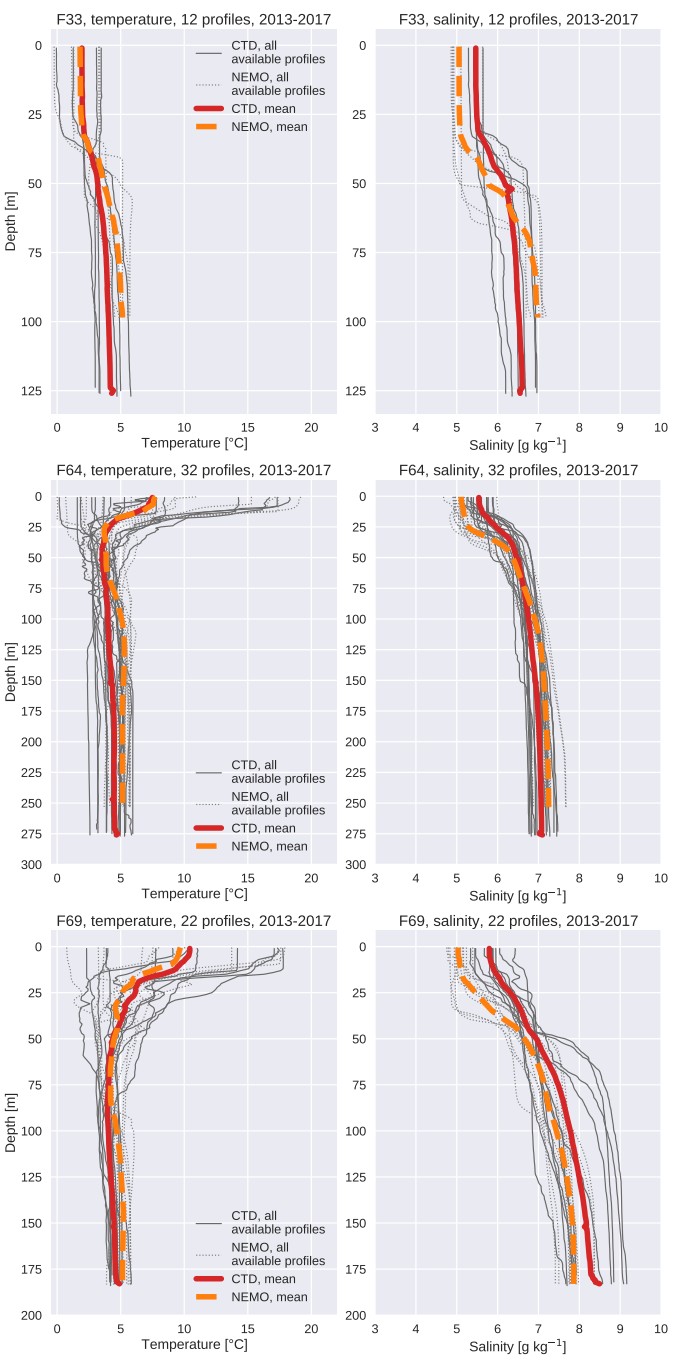

**Figure 3.** Temperature and salinity profiles from three monitoring stations 2013–2017. All available measurement profiles and corresponding modelled profiles are shown, along with their means. Please note that the number of profiles varies from station to station.

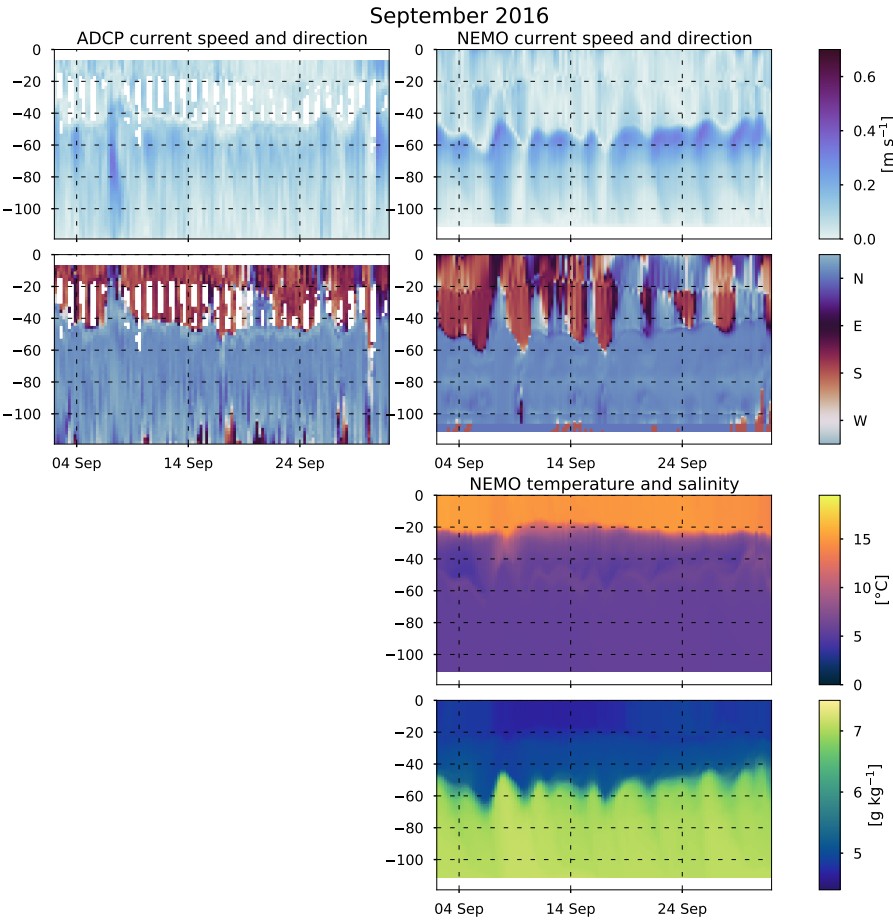

**Figure 4.** Current magnitudes and directions in September 2016 as measured at the ADCP station and as seen by the model. Also modelled temperature and salinity profiles are displayed.

## 3.2 Analysis of currents

### 3.2.1 Simulated and measured currents in the Southern Quark

Information about the quality of modelled currents is relevant when estimating the reliability of volume transport calculations.
Therefore, we compared modelled current magnitudes and directions with observations at the ADCP installation (location shown in Fig. 1). Both model and ADCP data were available for the period of August 2016–December 2017. While time and space averaged currents saved from the model often represent different things than the very local observations from ADCP instruments, they can still be used to get an overview of the model performance in the area where the ADCP was located.

The vertical profiles of the modelled currents have a similar structure as the measured currents. During autumn and winter,
there are two layers: an upper layer above the halocline and a deep layer below the halocline (Fig. 4). During the thermally





stratified period, three different layers are visible: a surface layer above the thermocline, an intermediate layer from the thermocline to the halocline, and a deep layer below the halocline. The surface layer is more pronounced in modelled current profiles than in measurements, as the ADCP data is missing the upmost 8 m layer. The depth of the thermocline in the model mainly undulates between 10–25 m and the depth of the halocline between 40–70 m (Fig. 4). ADCP current speed and direction

data shows that observed halocline depth is mainly between 40–60 m. As we consider mainly current dynamics and transport analysis, the changes of properties occurring at the halocline are more significant than those at the thermocline, for example regarding the direction distribution of currents. For this reason, we use "upper" or "surface layer" to signify the layer above the halocline and "lower" or "deep layer" the layer below it. We specifically state each time when the intermediate layer is being considered.

Modelled current magnitudes are highest in the surface layer of few meters, and below the halocline, at depths of 50–70 m. The strongest currents occur typically in late autumn or winter. The monthly mean current magnitudes vary between 0.05 m s$^{-1}$ and 0.20 m s$^{-1}$ in the upmost 10 m surface layer, and between 0.05 m s$^{-1}$ and 0.10 m s$^{-1}$ from around 10 m depth down to the halocline. Below the halocline, at the depths of approximately 50–70 m, the monthly means vary between 0.11 m s$^{-1}$ and 0.23 m s$^{-1}$. Below the depths of 70–80 m, there are less seasonal variations in current magnitudes, monthly means

being approximately 0.11 m s$^{-1}$.

To validate the modelled currents, we calculated monthly bias in the modelled horizontal current components and the current magnitude for the period of August 2016–December 2017. In general, in the upper 40 m layer the model under-estimated the U (zonal) component and over-estimated the V (meridional) component, and this led to a slight under-estimation of the current magnitude (Fig. 6). In the water column from 80 m downwards, the model over-estimated the U component and under-estimated

the V component, leading to under-estimation of the current magnitude. The largest monthly biases in the current magnitude, up to 0.11 m s$^{-1}$ at 60 m depth in February 2017, are seen in the layer between 50 m and 75 m, mainly due to over-estimation of the V component. During the summer months in 2017 (from April to August), the model under-estimated the V component in the 40–50 m layer by 0.005–0.043 m s$^{-1}$ and over-estimated the U component by 0.006–0.34 m s$^{-1}$.

Both in measured and modelled currents, the dominant current direction is towards the southerly sector (SW–SE) in the upper

layer and towards northerly sector (NW–NE) in the lower layer. However, the currents in the upper layer have more variation in direction, and also northerly currents occur. In the model, these northerly currents are vertically more uniform and last for longer periods of time than in the measurements. Moreover, the model shows northerly currents at times when the observed current direction is southward. For example at 10 m depth, the measured current directions were mainly towards south and south-east and the fraction of northward currents (towards sectors NW–NE) was small, 11 % of the whole comparison period

August 2016–December 2017 (Fig. 5). The prevailing current direction in the modelled currents was towards south, but the fraction of northward currents was larger than in the measurements, 26 % of the whole comparison period.

In the lower layer, there was less variation in the current directions than in the upper layer. For example at 70 m depth, the measured as well as the modelled currents were mainly directed towards north-north-west, but the modelled currents showed larger fraction of currents over 0.20 m s$^{-1}$ than the observations (Fig. 5). At 100 m depth, the observed currents were directed

towards north-west and north-north-west whereas the modelled current direction was dominantly towards north-north-west.





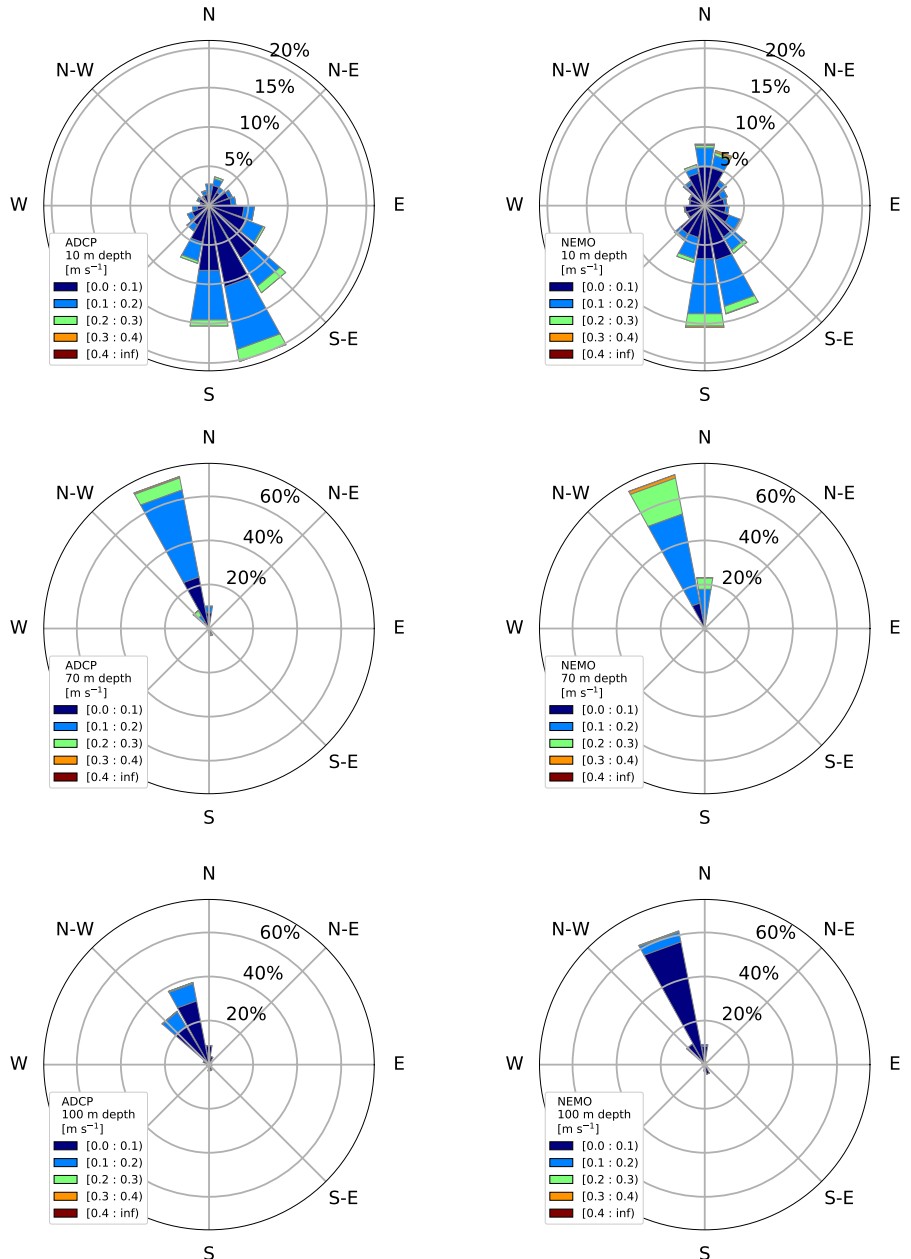

**Figure 5.** Comparison between measured and modelled currents at the ADCP location. Current roses are shown for 6 August 2016–28 December 2017 at depths of 10 m, 70 m and 100 m.

As the current direction at this depth follows the bottom topography, this difference between observed and modelled current direction is possibly caused by small differences between the real bottom topography and the model bathymetry.



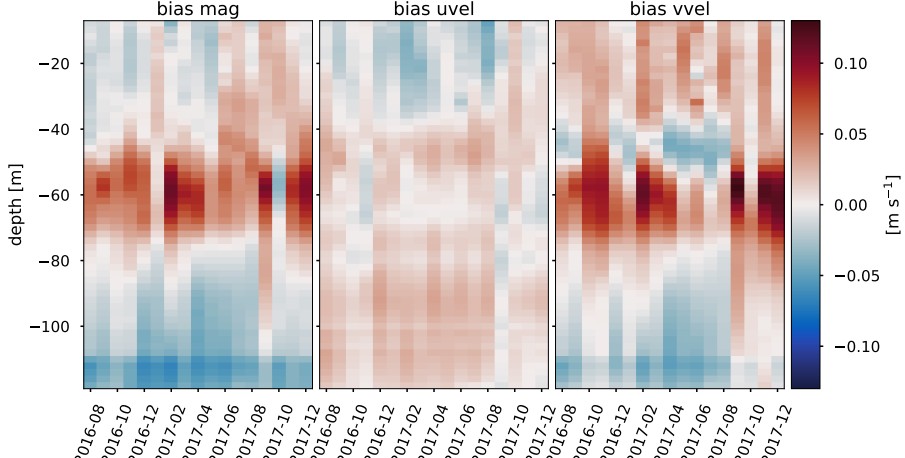

**Figure 6.** Monthly bias in the modelled current magnitude, and the U (zonal) and V (meridional) components.

### 3.2.2 Seasonality of currents in the Åland Sea

To study the seasonality of circulation patterns both in the surface layer and deeper in the water column, we calculated the
mean current field over the whole modelling period of 2013–2017 (Fig. 7) as well as the seasonal mean current fields (Fig. 8).
Winter was defined to be from December of the previous year to February (DJF), spring from March to May (MAM), summer
from June to August (JJA) and autumn from September to November (SON).

All the seasonal means show southward surface currents through the Åland Sea turning south-eastward or eastward in the
southern part of the Åland Sea (Fig. 8). In the eastern side of the Åland Sea, a characteristic feature is a counter-clockwise loop
existing in autumn, winter and spring. Its location has both seasonal and inter-annual variation. In general, the mean current
speeds are stronger at the western side than at the eastern side of the basin, as could be expected.

Autumn was the season with the largest inter-annual variation in current directions, but the southward and south-eastward
currents were still dominant in most years. In summer and spring, there was very little year-to-year directional variation. To
put it differently, the persistency of surface currents was significantly lower in the autumn than in other seasons. This is visible
in the lengths of the vectors in the seasonal mean figure. Although current magnitudes were not significantly smaller in the
autumn, the mean current vectors are still shorter just because the averaging of current vectors is performed component-wise.

While generally most seasons had southward currents, winter 2013–2014 was somewhat exceptional. The surface currents
were northward or north-eastward almost in the whole model domain. Of the other winters, the southward currents were
strongest in 2016–2017.

Deep layer currents show less seasonal variation in direction than surface currents and the current direction is generally
northward in all seasons. Figure 7 shows the mean circulation at 50 m depth as an example. In the southern part of the model
domain, the mean current direction is northward and north-westward along the channel that leads to the Lågskär Deep. In the
Åland Sea proper, there is a counter-clockwise loop covering the whole basin, with stronger currents on the eastern side of the

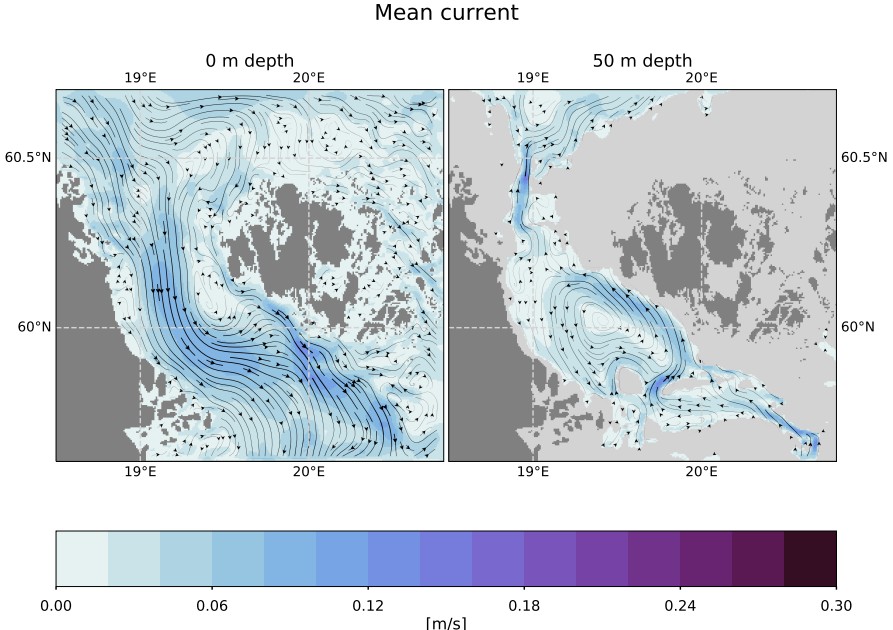

**Figure 7.** Modelled mean currents at the surface and at 50 m depth in 2013–2017.

basin. In the northern part of the basin, the currents continue northward along the Southern Quark, and the bathymetry steers
the currents north-eastward near the northern edge of the model domain. The persistency of the current direction is high in the
narrow channels and also higher on the eastern than on the western side of the loop in Åland Sea proper.

### 3.3 Volume transports in the Åland Sea

To better understand water exchange in the Åland Sea, we analyzed volume transports along six zonal (west-east) sections
across the basin (locations shown in Fig. 1). For this analysis transports were integrated over the upper part of the water
column down to 40 m depth, over the lower part of the water column below 40 m depth, and over the whole water column.
This roughly splits the water column to upper (and intermediate) layer above the halocline and deep layer below the halocline
(cf. Sec. 3.2.1). While the depth of the halocline varies somewhat spatially and temporally, the salinity profiles and ADCP data
suggest that 40 m is a reasonable estimate for this analysis. As our main interest is in the sub-halocline deep water transports,
and as analysis of currents revealed some significant current speeds in the model layers just below the halocline, we deemed it
was more important to choose a separating depth for the analysis that, for the most part, was either at the halocline or slightly
above it.

The locations of the transects (see Fig. 1) were chosen to support the aim to study especially the deep layer transports.
Starting from the north, the two northernmost transects were set north from Märket and Understen and close to them on both
sides of the sill to capture fluxes across the Northern Åland Sea sill. The third and fourth transects represent approximately

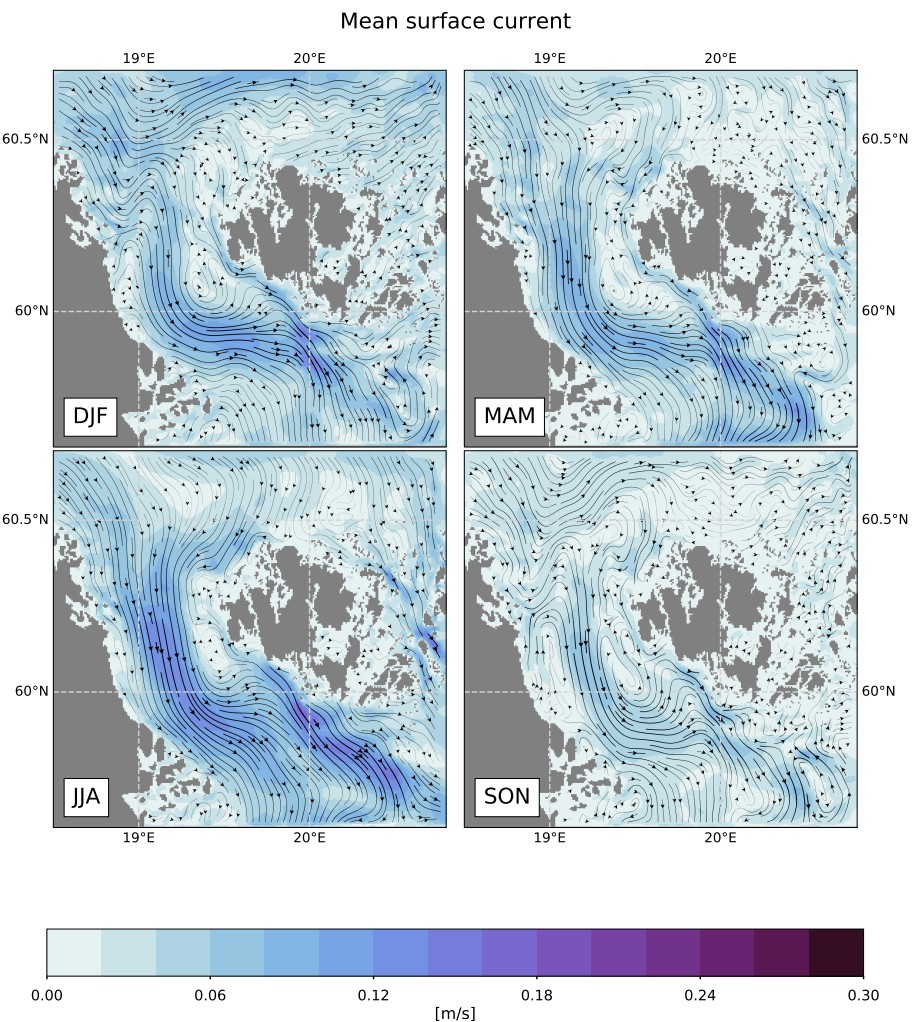

**Figure 8.** Seasonal means of modelled surface currents in 2013–2017. Please note that the means of current vectors are calculated as the means of elements in the vector, not as means of magnitudes. This means that in autumn (SON) when current stability is low, magnitudes of mean vectors (shown) are much smaller than means of magnitudes (not shown).





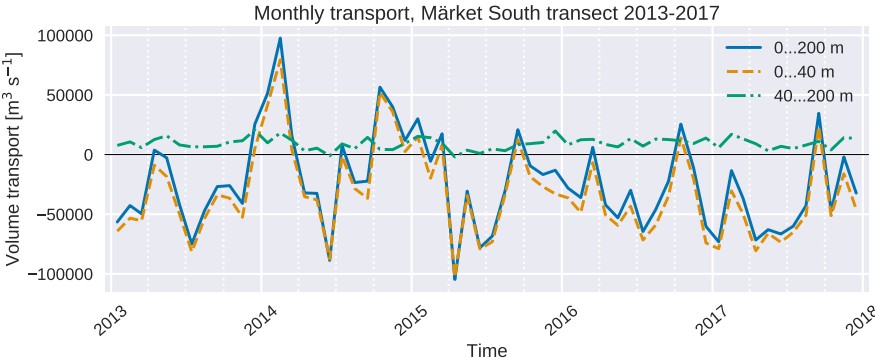

**Figure 9.** Time series of modelled monthly mean volume transports through the Märket South transect near the northern edge of Åland Sea 2013–2017. Positive values northwards. Values for the whole water column, the upper water column up to 40 m and the lower water column below 40 m are shown.

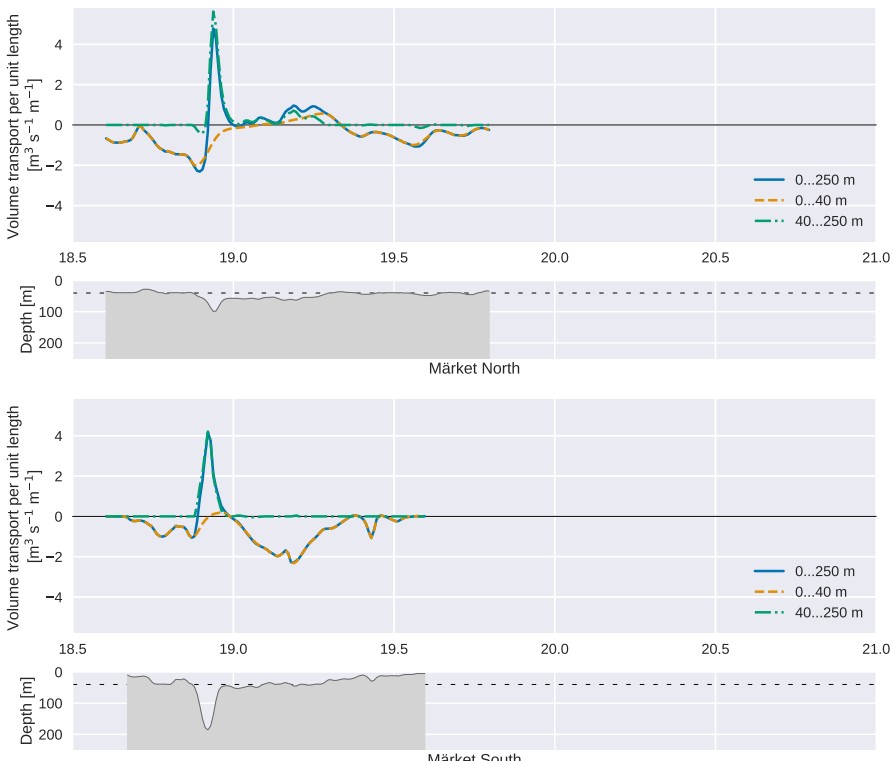

**Figure 10.** Volume transports per unit length along a transect, integrated over depth, from two northernmost longitudinal transects in the Åland Sea (MN and MS). Mean values for 2013–2017 shown. Positive values northwards. Values for the whole water column, the upper water column up to 40 m and the lower water column below 40 m are shown. Also, for each transect, a depth profile along the transect is displayed, with the 40 m threshold marked with a dashed line. For reference, one degree longitude is here approximately 55 km long.


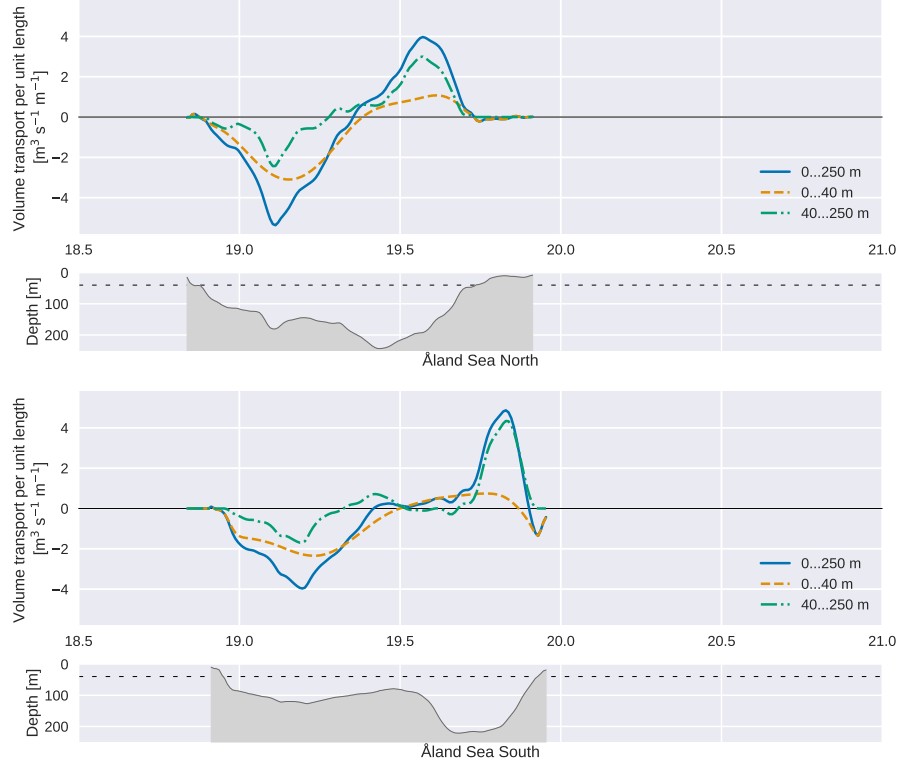

**Figure 11.** Same as Fig. 10, but for the two transects in the middle part of the Åland Sea (ÅN and ÅS).

the middle and southern part of the Åland Sea proper to explain the internal dynamics of the Åland Sea. The fifth and sixth
transects are located at the northern and southern edge of the Lågskär Deep to investigate transports in and out of this area in
order to explain the role of the Lågskär Deep in the water exchange and eventually on the mixing processes of waters coming
from the Baltic Proper.

A time series of monthly mean volume transports integrated over the whole transect is shown for the Märket South transect
in Fig. 9. While this time series plot is shown here for one transect only, the overall appearance of the time series with
corresponding averaging is very similar also for the other transects, although the quantitative values vary somewhat.

An overview of the modelled transports is consistent with the general knowledge of the water exchange in the area. The
deep layer transport goes to the north and the upper layer transport goes on average to the south. Transport in the upper layer is
much larger than in the lower layer and it dominates the integrated transport. This can be expected, as water entering the Gulf
of Bothnia from the south must at some point also exit the gulf towards the south. Furthermore, fresh river runoffs into the Gulf
of Bothnia leave the basin in the surface waters, further increasing the surface transport towards the south. There are, however,
months when the overall transport is towards the north, most notably in the autumns. The largest values of water transport


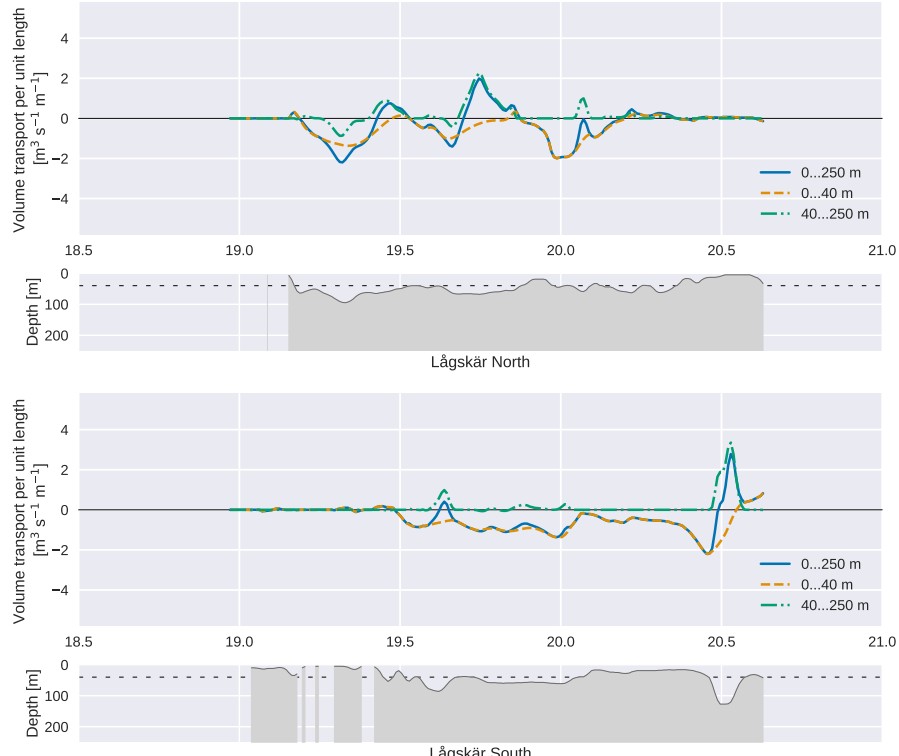

**Figure 12.** Same as Fig. 10, but for the two southernmost transects (LN and LS).

towards the south occur in late spring and early summer. Variability from one month to another and one year to another is significant.

Next, to illustrate the pathways of water, we calculated volume transports per unit length along these six sections (Figs. 10, 11 and 12). Mean values were calculated for the whole study period. Starting now from the south and moving towards north, waters enters the deep layer in the Lågskär Deep from the Baltic Sea Proper mainly across the Southern Åland Sill through the passage starting east of Bogskär (located south of the southern edge of our domain) and connecting to the south-east corner of the Lågskär Deep. The water then is transported to Åland Sea proper mainly over the Åland Sea Middle Sill between Söderarm

and Lågskär.

In the Åland Sea proper, we see a structure where waters flow northwards in the eastern side of the basin and southwards in the western side of the basin. Unlike for the other parts of the Åland Sea, in this case there is also clear southwards transport in the lower part of the water column in the western part of the transect. This in line with the loop visible in this area in Fig. 7. As the eastern part of the basin is much deeper than the western part, also the transport below 40 m is much larger in the east than

in the west. Finally, for the northernmost parts we see deep water flowing northwards between Understen and Märket through the Southern Quark Strait.





These transects also reveal some interesting details about the transports. For example, in the southernmost transect we see that, in addition to the strait located at approximately 20.5° E, some deep water enters the basin also through the depression at 19.6° E at the south-west corner of the Lågskär Deep. The primary route is on average responsible for 75 % of the transport. The rest flow through the western parts of the transect.

Furthermore, at the northern edge of the Lågskär Deep, the deep water transport is divided through three different routes. Transport through the passage east of Lågskär at approximately 20.1° E seems minor when compared to the transport west of Lågskär and east of Söderarm between 19.3°and 20.0° E. Between Lågskär and Söderarm, deep water can take two different routes, and our model indicates that the majority of flow takes places through the eastern route at approximately 19.7° E.

## 4    Discussion

We built the modelling configuration presented here in part to address some of the issues discovered by Tuomi et al. (2018) and Miettunen et al. (2020) when modelling the Archipelago Sea. An obvious improvement is the inclusion of the Åland Sea in the model domain, which made it possible to study water exchange is this area. This is the first study of these exchange processes in such high resolution. Our results are mostly consistent with earlier understanding, with some new details and insights of water exchange processes in the area. The results presented herein are useful for purposes such as planning of future ocean observation and marine monitoring activities.

### 4.1    Currents and circulation

Investigation of currents revealed that the model could capture the overall layered structure of currents reasonably well and the model results were plausible. The two-layer structure of currents (or sometimes even three-layer, when a seasonal thermocline was present) observed in the ADCP measurements was well represented in the modelled currents. The model was also able to represent the depth of the halocline separating these two layers quite reliably.

There are some instances where direction distribution of the modelled currents somewhat differed from the observations at the ADCP station. Notably, the model showed a larger fraction of relatively low-speed northward currents in the upper layers at the ADCP measurement point than was actually observed. It is worth discussing this difference briefly, as it is related to the reliability of transport estimates in the upper layer. It seems that this difference can in large part result from the inevitable inaccuracies in the placement or timing of submesoscale features in the model. Our model data shows that the circulation field in the Southern Quark area is spatially highly variable with a number of eddies and vortices. Small differences in the locations with submesoscale eddies, for example, can result in significant differences in the direction distribution of currents at a single point, if that point ends up on the opposite sides of a current loop in the model and in nature.

Our investigation revealed that in our dataset almost all cases where there were northwards currents in the model and some different direction in the observations were times of turbulent and variable circulation field in the model and relatively low current speeds in the vicinity of the ADCP site. Quite often the model current field showed a relatively strong meandering northward mesoscale current east of the ADCP site and an abundance of short-lived submesoscale structures with lower current





speeds on its either side that could perhaps be described as "submesoscale soup" (McWilliams, 2019). The model also showed
many cases where upper layer northward currents were modelled well.

Conclusive investigation of these differences would require better spatial coverage of ADCP measurements. For our study period, we only have at our disposal data from one ADCP station in this area. The location of this ADCP was selected to measure currents in the deep layer in the passage between Åland Sea and Bothnian Sea, but for validating surface layer currents a southward location would be better.

Some other sources also support the notion that incorrectly placed submesoscale features and spatial variability might be a significant contributor to these kinds of differences in the direction distribution. For instance, Ehlin and Ambjörn (1977) measured currents at four stations in the Southern Quark. Their current measurement stations were located a couple of kilometres apart. They found that, at times, current directions could be completely opposite from one station to another, indicating similar spatial variability we saw in our model results. It is also prudent to point out that the boundary of our model domain is relatively
close to the Southern Quark area. As this is the case, it is plausible that any inaccuracies – even small ones – in the boundary condition data to be reflected in the locations of eddies in the model. Also any inaccuracies in wind forcing can be significant.

It is also worth mentioning that gaps in the ADCP measurement record can complicate their interpretation. It is plausible that some northward currents could have gone unrecorded. On average, 33 % of all ADCP measurements are missing in the surface layer, with some months lacking up to 77 % of measurements due to measurement difficulties. However, based on our
data we estimate that northward currents have not disproportionately gone unrecorded and this is not a major contributor to this issue.

Investigation of modelled seasonal surface circulation patterns in the Åland Sea revealed an overall structure where southwards currents could be observed throughout the Åland Sea with strongest currents along the western edge of the basin. The magnitude of this current varied from one season to another, but the direction was more or less the same. As far as the other
parts of the study area are concerned, there is more variability in the direction of the mean current near the southern and northern edge of the area. There were also two cases during the investigation period, winter 2013–2014 and autumn 2014, when the overall direction to the mean seasonal current was towards the north. As expected, near-bottom currents were much less volatile and had less variability than surface currents.

While seasonal means are useful for a number of applications, care should be taken when they are applied. It is important,
for example, to remind ourselves that there is a lot of variability in the circulation patterns in shorter time scales that is hidden by the averaging process (cf. Westerlund, 2018). Near-surface circulation patterns are especially affected by wind forcing, for instance. Averaging current vectors for seasons with lower persistency, namely autumn, results in much lower mean speeds than averaging current magnitudes with no consideration for their direction.

## 4.2  Volume transports

The overall picture of modelled water exchange in the Åland Sea mostly followed what has been reported earlier. More saline water from the Baltic Proper enters Åland Sea mostly through Lågskär Deep, after which it is transported through Åland Sea proper northwards, ultimately reaching the Bothnian Sea through the Southern Quark. However, it was interesting that in our





model only 75 % of the water that enters from the Baltic Proper in the deep layer is transported through the primary route in the south-east corner of Lågskär Deep while a significant percentage bypasses the Lågskär Deep entirely from its western side.

It is somewhat cumbersome to find an appropriate frame of reference for our water transport results from the literature. Previous estimates of water exchange through the Åland Sea and the Archipelago Sea have often employed a Knudsen type budget approach. As Myrberg and Andrejev (2006) notes, there is significant variance between the results of different studies depending on factors such as averaging period, temporal coverage of measurements and location of transects in relation to dynamical features. Furthermore, these estimates are typically for the whole Gulf of Bothnia, while we concentrate on the

Åland Sea part and leave the Archipelago Sea for future studies. Therefore, as the water exchange through the Archipelago Sea is not included in our estimates, we should look at these previous results more as upper bounds. Myrberg and Andrejev (2006) point out that these estimates can nevertheless be compared to modelling results as a first approximation.

Ambjörn and Gidhagen (1979) give estimates for net water transport in the Åland Sea. Values for monthly transports for a few months in the late seventies are reproduced in Table 1. These numbers are based on current measurements and empirical

orthogonal functions (EOF). While there is notable inter-annual variability in transports and these values cannot be directly compared to our results, looking at our Fig. 9, we see that the direction (southwards) is the same, as is the general magnitude in the latter half of the year (approximately from zero to $-10^5$ m$^3$ s$^{-1}$).

**Table 1.** Net water transport in the Åland Sea according to Ambjörn and Gidhagen (1979). Last column calculated assuming 30.437 days per month.

| Date | Net transport in km$^3$ per month | Net transport in m$^3$ s$^{-1}$ |
| --- | --- | --- |
| 1974 August | -139 | -52900 |
| 1974 September | -27 | -10300 |
| 1974 October | -141 | -53600 |
| 1974 November | -90 | -34200 |
| 1977 June | -15 | -5700 |
| 1977 July | -87 | -33100 |
| 1977 August | -89 | -33800 |

Also Ehlin and Ambjörn (1977) published estimates of water transport to the Gulf of Bothnia, partially based on same data as Ambjörn and Gidhagen (1979). They used tide gauge data from the Gulf of Bothnia, and current measurements from several

stations in the Understen-Märket area in 1973–1974. When they investigated daily mean transports in the area, they arrived at values which varied mostly between 5 and 10 km$^3$ d$^{-1}$, which translates approximately to 58000–120000 m$^3$ s$^{-1}$. They also saw much higher values, which is expected as they recorded daily transports.

Our modelled values from the same area in Fig. 9 mostly fall within the range of values given by both Ambjörn and Gidhagen (1979) and Ehlin and Ambjörn (1977). This builds confidence for using model approach in these type of studies.

The veracity of modelled transports depends heavily on how well the model captures magnitudes and directions of current fields. The ADCP validation suggests that modelled currents are mostly trustworthy, at least near the ADCP station. This in turn





would suggest that the overall magnitude of transports could be sensible. As discussed, the model seems to indicate a somewhat greater fraction of northwards currents in the surface layer than is present in observations, which might mean that in some cases the surface layer volume transport would perhaps be overestimated. However, examination of such cases revealed that current magnitudes were mostly moderate. Cases where the model interpreted the dynamical situation completely incorrectly seem to be very rare. In the time period when ADCP observations were available, August 2016–December 2017, the most notable case was in October 2016. As in this month we had stronger northward currents in the model than in the ADCP data, it suggests that we should treat the positive value for surface layer volume transport for that month in Fig. 9 as an upper limit. While it stands to reason that the modelled upper layer transport may overall be somewhat more uncertain than in the deep layer, other differences in this validation were less major. If ADCP measurements with better spatial and temporal coverage became available, they could clarify this issue further. Also it would be useful if future current measurements could reliably capture currents in the whole water column even in deeper areas.

### 4.3 Model configuration and parameterizations

Because of the diverse bathymetric and hydrographic features of our study area, one of the key challenges for this study was finding the right balance between model stability and mixing. A notable difference to the configuration by Tuomi et al. (2018) was our choice of the z* vertical coordinate system instead of $\sigma$ coordinates.

While the $\sigma$ coordinate system is highly popular in coastal modelling applications, it has some issues that make it less than ideal for the Åland Sea – Archipelago Sea area. One of these issues is the presence of the internal pressure gradient error. It can be especially problematic for coastal problems that include steep bathymetric gradients such as canyons or seamounts (Fringer et al., 2019). For many coastal problems, where strong tidal currents and mixing dominate, the internal pressure gradient error is less of an issue. But the Baltic Sea is micro-tidal, so for our study area this issue is potentially bothersome. Indeed, Tuomi et al. (2018) found significant over-mixing especially in the deeper channels of the Archipelago Sea.

One way to address this issue is to use geopotential coordinate systems, which do not exhibit the pressure gradient error. Unfortunately, these systems have their own problems. For example, the standard geopotential vertical coordinate system limits the size of the topmost vertical level, which makes it more difficult to study near-surface dynamics (Klingbeil et al., 2018). This issue can be resolved by the use of the z* system, which allows finer vertical resolution.

Considerable amount of manual work was required to ensure that bathymetric features of the area were represented in the model grid as accurately as possible, so that instabilities were not introduced by bathymetric artefacts appearing due to the limited grid resolution. At the same time, we had to find values for mixing parameters that produced sensible results while simultaneously being high enough to maintain model stability.

The end result is satisfactory in the sense that this model configuration seems to be able to reproduce even steeper gradients quite well and does not suffer from spurious over-mixing in the same extent as the configuration by Tuomi et al. (2018). Still, it is evident that further tuning of model bathymetry, bottom friction and mixing parameters would be beneficial to improve results further. Depending on the scales and phenomena that are investigated with this configuration in the future, also higher resolution forcing data might be useful.





One limiting factor for model accuracy is that high-resolution bathymetric data from the area either does not exist or is not generally available. Sometimes, the bathymetry simply hasn't been measured with high enough accuracy, while in other cases the availability of existing data for scientific study is limited by non-scientific factors such as national security concerns or commercial interests. In the future, efforts to make higher resolution bathymetric data available would make further model
improvements possible.

### 4.4 Outlook

This study lays groundwork for further studies that could shed light on the open questions relating to water exchange between Baltic Proper and Gulf of Bothnia. For example, there is still uncertainty regarding the routes and volumes of water in the Archipelago Sea. Water exchange could be investigated on both longer and shorter time spans and the role of significant water
exchange events could be elaborated. Also, salinity transports are interesting when working to understand connections to the environmental changes in the Bothnian Sea.

Another possible application for volume transports computed from this configuration could be to compare them to results from a coarser configuration. Many regional Baltic Sea models are unable to fully resolve the Åland Sea area due to limited resolution. State-of-the-art regional configurations nowadays typically have horizontal resolutions of around 1 NM (see e.g.
Kärnä et al., 2021). Efforts to develop regional configurations further might benefit from such analysis.

Further developments of the modelling configuration could improve its accuracy. For example, boundary conditions can have a major impact on the results. For instance, salinity biases present in the boundary conditions can be quickly propagated to the whole modelling area. One possible way to address these issues could be the development of an improved configuration for the area with two-way nesting.

In addition to water exchange and transports, this modelling configuration could also be used to investigate a number of other topics, some of which we mention here. Questions related to the environmental health of the sea and nutrient reductions could be studied. This configuration could, for example, provide current fields to nutrient load modelling in a similar manner as a COHERENS based model configuration was used by Lignell et al. (2018). The relatively high resolution could also allow studies of coastal processes with detail and spatial coverage that so far has not been possible. Furthermore, this configuration might
prove useful for assisting marine spatial planning, for example by providing data for studies of connectivity of marine habitats. Also, substance transport modelling is a topical issue that could be supported with this setup. Both Lagrangian and Eulerian transport studies could be interesting, and could be conducted either by adding an online component to this configuration, or by offline coupling another model for substance transport in a similar manner as Miettunen et al. (2020).

### 5 Conclusions

We studied volume transports through the Åland Sea, Baltic Sea with a new high-resolution hydrodynamic model configuration.
Investigation of modelled current magnitudes and distribution in the area provided encouraging results regarding the ability of our configuration to capture the overall dynamics and volume transports in the Åland Sea. We found that modelled circulation





patterns in the study area were variable. Currents typically had a two-layer structure separated at the halocline. Seasonal means revealed that there commonly was a southward current in the surface layers in all seasons. The stability of currents was notably

lower in the autumn compared to other seasons. In the deeper layer, currents were directed by bathymetric features and mostly towards the north for all seasons.

Analysis of modelled volume transports showed how deep water is transported northward from the Baltic Proper to the Bothnian Sea over the sills and via the available passages. Time series of volume transports from northern Åland Sea revealed that monthly averages of deep transport were consistently towards the north. On the surface, the net transport was towards the

south. However, in most years, there were months in late summer or early autumn with northward monthly mean transports in the surface layer.

Our analysis indicates that the dynamics in Lågskär Deep are more complex than has previously been thought. It seems that while Lågskär Deep is the primary route of deep water exchange, still a significant volume of deep water enters the Åland Sea through the depression west of the Lågskär Deep. The primary route is on average responsible for 75 % of the transport, while

the rest flow through the western parts of the transect.

In future studies, the reliability of current and transport estimates could be improved with increased spatial and temporal coverage of current observations from this area. While the configuration reproduced the overall temperature and salinity dynamics and sea surface heights in the area in an adequate manner, model validation would further benefit from CTD datasets with good spatial and temporal coverage. High-quality forcing and boundary condition datasets would also help to build further

confidence in the water exchange estimates.

*Code and data availability.* The standard NEMO model source code is available from the NEMO web site at https://www.nemo-ocean.eu/. The NEMO configuration files for the Åland Sea and Archipelago Sea setup are available from https://github.com/fmidev/nemo-archs. Model boundary condition data are available from Copernicus Marine Service at https://marine.copernicus.eu/. Atmospheric forcing data are available from Copernicus Climate Service at climate.copernicus.eu. The bathymetric input file for the Åland Sea and Archipelago

Sea NEMO configuration is not available due to current SYKE policy regarding the VELMU bathymetric data. River runoff data are available from SYKE on request. Föglö Degerby SSH data are available from the Finnish Meteorological Institute (FMI), see https://en.ilmatieteenlaitos.fi/open-data. The ADCP dataset for the station in the Southern Quark is available on request from the authors. CTD monitoring data are provided by the Finnish Environment Institute (SYKE) and Centres for Economic Development, Transport and the Environment (ELY), see http://www.syke.fi/en-US/Open_information.

*Author contributions.* AW, EM and LT designed the modelling configuration, with input from PA. The configuration was implemented by AW with major contributions from EM. All authors contributed to the design of the numerical experiments, which AW then carried out. AW and EM validated the model results and performed visualization. EM was responsible for the analysis of currents while AW was responsible for transport analysis. All authors took part in all analyses and discussion of the results. AW and EM prepared the manuscript with major contributions from all co-authors.





*Competing interests.*  The authors declare that they have no conflict of interest.

*Acknowledgements.*  The authors would like to thank Hedi Kanarik for processing the ADCP data and lending her expertise in their interpre-
tation.

This work has been partially funded by the Strategic Research Council at the Academy of Finland (contract number 312650, BlueAdapt),
and Finnish Ministry for the Environment Water Protection Programme 2019–2023.

This study uses data from the Baltic Sea Bathymetry Database (Baltic Sea Hydrographic Commission, 2013) version 0.9.3, downloaded
from http://data.bshc.pro/ on 24 July 2018. This study has been conducted using E.U. Copernicus Marine Service Information. Contains
modified Copernicus Climate Change Service information 2020. Neither the European Commission nor ECMWF is responsible for any use
that may be made of the Copernicus information or data it contains.





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
