# Peer review of "Refined estimates of water transport through the Åland Sea, Baltic Sea"

_Ocean Science, 2021_

## Author Comment (AC1)

**Author's response to RC1**

Westerlund, A., Miettunen, E., Tuomi, L., and Alenius, P.: Refined estimates of water transport through the Åland Sea, Baltic Sea, Ocean Sci. Discuss. [preprint], https://doi.org/10.5194/os-2021-56, in review, 2021

Below, reviewer comments are displayed with a gray background, while author responses are without highlighting.

**RC1**: Anonymous Referee #1, 28 Jun 2021**

> Review of submission os-2021-56

> Title: Refined estimates of water transport through the Aland Sea, Baltic Sea

> Authors: A. Westerlund, E. Miettunen, L. Tuomi, P. Alenius

> General:

> The study by Westerlund et al. investigates main pathways of water transport through the Aland Sea by means of a high-resolution regional model application. Previous observational and model approaches provided limited understanding of the transport and circulation structure in that area because of the strong seasonality of the regional atmospheric forcing as well as the complex bottom topography which requires high-resolution data coverage both in space and time to adequately capture the main transport characteristics. The aim of the study, as I understand it, is to gain insight into exchange dynamics between the Baltic Proper and the Bothnian Sea at interannual and seasonal scales of the recent past. To this end, the authors constructed a bathymetric representation of the Aland Sea with an unprecedented resolution of 500m horizontally and 200 vertical layers. The applied hindcast simulation with NEMO provided hourly to daily model output for the period 2013-2017.

> While the model setup has been evaluated against available station data and seems to perform sufficiently well, my main concern is related to the rather limited use of the comprehensive model output. Substantial parts of the ms are dedicated to the discussion of the model biases and their possible origins. The transport dynamics as being the main focus of the study, by contrast, are presented in a rather descriptive way whithout analyzing and discussing any driving mechanisms or broader context that would finally gain our process understanding of the Aland Sea circulation. Apart from the shown figures, the transport rates are not even quanitfied. In this way, the only new finding of the study seems to be that about 25% of the transport entering the Aland Sea from the south does not follow the main strait at 20.5°E but rather happens through a topographic depression at 19.6°E. Of course, it is valuable to reflect on the model biases. But if the focus on the analysis and discussion of the model results is underrepresented, it conveys a rather defensive and repetitive flavor.

> Nevertheless, I do see great potential to use the performed simulation for further analysis that, in my opinion, would substantially increase the depth, relevance and impact of this

*study. For instance, questions that naturally arise while reading the present version of the ms and could well be addressed, are: What drives the occasional northward turn of the surface flow? Is it exceptional wind conditions? Does the northward surface flow lead to SSH and pressure anomalies in the Aland and Bothnian Sea? Do these anomalies temporally weaken the more steady sub-halocline northward flow? What drives the sub-halocline gyre in the Aland Sea Proper? Why is the surface circulation strongest in summer? Is this related to melt water discharge from land? What are the transport budgets of the individual basins of the Aland Sea? Is the sub-halocline northward transport a continuous steady flow or is part of the inflow returned and exported southward horizontally or temporally or via mixing with the surface flow? Are there regionalized future projections for wind and freshwater discharge available that could be utilized to hypothesize on potential climate change impacts (e.g. those used by Meier et al. 2021, https://doi.org/10.1038/s43247-021-00115-9)? Could one also hypothesize/extrapolate from your results on the water mass exchange to the nutrient supply into the Bothnian Sea?*

*> The conclusions could then follow a more explanatory line, if supported by the model results, such as: Continuous northward transport into the Bothnian Sea is dominated almost entirely by sub-halocline water masses. Northward flow potentially occurs in any area where the bathymetry exceeds the local depth of the halocline. If these areas are wide enough (with respect to the deformation radius?) the Coriolis force aligns the northward flow to the eastern side of the passages, which leads to a shoaling of the halocline at the eastern side and a deepening at the western side. Wind conditions can drive the predominantly southward surface flow of the estuarine-type general circulation towards the north on monthly to seasonal scales, causing significant anomalies in SSH and halocline structure. Further conclusions could be drawn from addressing some of the questions given above.*

*> I therefore would like to encourage the authors to dive deeper into the subject, provide more detailed analysis of the simulation already available, and from this derive more comprehensive and thoughtful conclusions.*

We would like to thank the reviewer for providing this insightful and encouraging commentary of our manuscript. It is highly appreciated.

The suggestions made by the referee regarding additional questions for analysis are excellent. We have carefully considered them and implemented a number of changes to the manuscript. We have expanded the analysis of wind conditions. We have revised how model biases are analyzed. We also added more information about the transport rates, as requested. We hope these modifications improve the depth and relevance of this study.

We have revised the introduction so that it answers questions about the scope and the objectives of the study better and more clearly. We have also clarified how this study sits in the larger research plan we have for the coming years. We think that these changes based on reviewer suggestions were beneficial and that the introduction has improved.

As far as we understand, the main criticism the referee is offering here is that the referee feels the manuscript falls short of the potential it could have. We find this point of view understandable. We are planning several further studies where this modelling setup is used and had to limit the scope of the already rather extensive current study somehow. We note

that the main focus of the paper, as set out in the title for example, is to provide refined estimates of transports through the Åland Sea, and this is done in the manuscript.

We agree with the reviewer that the discussion of model biases in the manuscript is somewhat more extensive than would have been the bare minimum. The reason behind this is that this manuscript is the first one where this particular model setup has been used. We felt it was important to discuss the quality of the model results in detail here to build confidence in further studies we have planned to conduct with this modelling setup. We are glad that the referee feels that this setup is performing sufficiently well.

We would like to thank the reviewer especially for raising in this review a number of interesting unanswered questions about water transport in the area. For example, the question regarding nutrient supply to the Bothnian Sea is certainly something that motivated our efforts as a long term goal and that warrants further study in upcoming years. Deeper analysis of transport dynamics is planned for further studies. While it is not possible to address all of these questions here given the time and scope constraints, we hope to continue on this line of research for many years to come, and to be able to resolve most of these questions. We highly appreciate the encouragement the referee is offering for further analysis of the data in this manuscript.

We hope the changes we have implemented address the concerns raised by the referee. We believe the manuscript has notably improved in this process. Please also find below our detailed answers to the specific comments made by the reviewer.

*Some minor specific comments:*

*> L21: What are these changes in the eutrophication status? Would be helpful information to better understand the context of the study.*

We have modified the introduction to more clearly discuss how eutrophication relates to this study.

*> Fig.1: Would be helpful to have an additional (small) inlet that shows the entire Baltic Sea and marks the location of the Aland Sea.*

We understand the point and in fact experimented with this modification already before initial submission. This Figure is somewhat challenging, as it needs to convey a lot of information regarding the main focus area, its location, geographical references and the area of the model domain. Also important are the readability of the map as well as that it is appropriate for the context. After a lot of experimentation, the decision was made to leave the Baltic Sea map out, as it was difficult to combine it with the two other maps in a readable style. We understand that this is a compromise, and while we would prefer to keep this map as it is, we are willing to experiment with different solutions, if it is considered important. Perhaps a new Figure for the Baltic Sea map on another page might be the most feasible solution.

*> L24-35: This paragraph would be better structured if it was split between the Aland Sea and the Archipelago Sea.*

Done.

> *L40: Maybe extend the last sentence by: " ... as the Archipelago Sea is too shallow to establish significant sub-halocline fluxes."*

We would prefer not to state this so categorically in this paper. Although the Archipelago Sea is very shallow, there still are relatively deep canyons and pathways there. While their role might be relatively minor in this respect, it would require further study to quantify it in sufficient detail.

> *L46/47: It is stated that there are 'no clearly defined water masses' in the Aland Sea but in the next sentence, 'the existence of a deep water type' is mentioned. Isn´t this contradictory?*

We agree with the reviewer, this is a bit confusing. Hela (1958) seems to agree as well. He described the use of a TS diagram here "schematic and more or less arbitrary" and stated: "The word 'water type' is used instead of the more traditional one of 'water mass' in order to emphasize their more or less varying character." We have modified the paragraph, hopefully clarifying what Hela meant.

> *L55: The unit [g kg-1] is used to refer to a change in salintiy. The cited study by Palosuo, however, dates back to the year 1964, where probably [psu] was used.*

The reviewer is of course correct that Palosuo did not use [g kg-1]. He used per mille as the unit of salinity, which in this case corresponds to the same value in g/kg. We used [g kg-1] for the benefit of the reader in this indirect reference to make this paragraph easier to read. It may be interesting to note here that psu was not yet defined in the 1960's. In 1960's "the salinity was determined by titration with silver nitrate solution according to the method by Knudsen, and using Copenhagen Normal Water as the standard."

> *L62-66: This paragraph can be condensed to: "... More recently, numerical modelling allows us to investigate intra- and inter-annual variability with much richer detail than we could with observations alone."*

The paragraph has been condensed. We kept the reference to spatial and temporal coverage, as this is discussed later on in the manuscript. Hopefully the paragraph is acceptable now.

> *L82: "... in this topographically complex and irregular area."*

Fixed.

> *L83: "... information about the bottom topography and related dynamics of different exchange pathways."*

We would like to keep the mention of sill depth here to emphasize its importance. We modified the sentence, hopefully it is more clear now.

> *L88: "... represents realistic bathymetric features in the investigated area."*

This change would slightly alter the intention of the sentence, but we did try to clarify the sentence. Hopefully it is better now.

> *L89/90: The aim of the study is not well outlined. I assume, the aim is at least to provide a detailed understanding of the present-day water mass exchange dynamics in the Aland Sea. And maybe some more such as to draw conclusions on the nutrient fluxes in the area or generally to provide information for the development of a science-based marine management strategy?*

We agree that the aim and scope of the study needed clarification. Hopefully our modifications have improved the situation.

> *L96: You may refer to the flushing times of the Aland Sea to justify the comparatively short spinup time of 6-7 months.*

We have added information about the water renewal time in our model to section 3.3.

> *L110: Why does the use of a sea ice model with thermodynamic formulation reduce computational demands?*

In our profiling of model code we have found that when a fully coupled ocean-sea ice model is in use, the dynamic solver in the current model code for sea ice dynamics can take several tens of percents of computational power required by the model. We think that in a study like the present one, the tradeoff of turning off the full dynamics of the sea ice field in favour of a longer run makes sense. As the modelling system is updated and developed in the future, we will continue to do profiling to investigate any changes.

> *L110-114: Would you expect that the water mass exchange is different during years/winters with large ice cover?*

We expect that if there was a severe ice winter during the modelling period, this might introduce additional and unnecessary uncertainties to the analysis.

> *L116-119: The calculation of volume transports does not have to be explained. These sentences could be condensed to: "We analyze volume transports across several transects to investigate the pathways of water exchange more closely."*

We in principle agree that this paragraph is perhaps almost overly detailed. However, based on our discussions with our colleagues before and during this study, we expect that this article will be of interest to relatively many oceanographers whose primary interest does not lie in physical oceanography. Based on these discussions, we believe it might be beneficial for these readers if we define here what we mean by volume transport. We would therefore prefer to keep this paragraph in the manuscript.

> *L128-133: Suggest to condense this paragraph to: "To mitigate artificial interpolation issues we checked and edited ... to ensure that it accurately represents the coastline and depth variations in the 0.25 NM resolution model domain."*

This modification would, in our opinion, make the paragraph somewhat incomplete, as then the reader would not be informed of the kinds of issues that typically result from the bathymetry compilation process. We believe it is important also to openly discuss challenges encountered during the study. We have, however, tried to condense the paragraph to address this comment. Hopefully it has improved.

> L134-140: Suggest to delte this paragraph as it does not contain important information.

We respectfully would prefer to keep this information. Our reasoning is that this is relevant for the reader so that they are able to evaluate and reflect which areas in the model domain are most challenging from the bathymetric point of view.

> L141: Why is it advantagous or necessary to smooth the steepest bathymetry gradients?

The matter of model tuning involves a compromise where the model is tuned to perform as realistically as possible, while at the same time keeping the model numerically stable enough so that the model runs are able to complete successfully. We modified this sentence to clarify this.

> L182: What are these processes? Would be helpful to name a few examples: "... other processes such as ..."

This sentence was somewhat incomplete in the submitted version. We have now clarified it.

> Fig.3: Have you compared the station data also with the ocean reanalysis product you use to drive the model? As you are mentioning in L457, might be that the S biases are related to the boundary conditions.

Good idea. We added the reanalysis product mean to Fig. 3 to illustrate the biases.

> L209ff: Suggest to delete the first sentence and start with "We evaluate modelled current magnitudes and ...". Also suggest to delete L211-213 (While time ... ADCP was located.)

We included this information on how ADCP validation was performed to inform the reader of the challenges involved with validating currents and to establish why this section was included. We believe this is not common knowledge with all of the intended audience of this manuscript. For this reason, we would prefer to keep this information.

> L261: Why can the stronger surface flow at the western side be expected? Would be interesting to elaborate more on the dynamics.

This fragment at the end of the sentence was likely a reference to earlier results in the literature concerning currents in the Åland Sea, which are not really relevant for this analysis. It seems that we have left this fragment in the submitted manuscript by mistake. We have therefore removed it. Thank you for pointing this out.

> L264: Why is the persistency of surface currents lower in autumn? Due to stronger or more frequent south-westerly wind conditions?

We have expanded the analysis of wind conditions in the manuscript. Hopefully it now addresses this comment.

> L267: What driving mechanism turned the surface flow northward during winter 2013/2014?

Hopefully the expanded wind analysis also addresses this comment.

> Fig.8: Why is the surface circulation strongest in summer? Due to melt water discharge and export?

This is likely related to the wind distribution in summer having a large percentage of relatively strong northerly winds supporting the transport of excess fresh water southward from the Bothnian Sea, but lower percentage of relatively strong S and SW winds than spring. Comprehensive analysis of this would require more time that was available at this point, so we left out this still somewhat speculative explanation from the manuscript. Recent analysis of Ferrybox data by Äijälä (2019), however, supports this assumption, showing lower salinities in the Bothnian Sea during summer months (their Fig. 3.10).

Reference: Äijälä, C. (2019). Suolaisuuden ja lämpötilan vaihtelu Pohjanlahdella perustuen FerryBox-dataan. http://urn.fi/URN:NBN:fi:hulib-201910083600. Master's thesis, University of Helsinki. In Finnish.

> L296: At least give correlation coefficients to support this statement.

Thank you for pointing this out, this comment made us realize our sentence was quite badly written and could be understood in English differently than was intended. The purpose of this sentence was only to say that quantitatively the values at different transects are of the same order, rather than make statements of time series correlation. However, we have now added more information about the other transects. Hopefully this paragraph is more clear now.

> L316: "... through the Northern Aland Sill."?

Exactly, over the sill to the strait. We rephrased it to make it more clear.

> L326: What are 'the issues discoverd by Tuomi et al. and Miettunen et al.'?

Unfortunately, we had written this sentence in a way that did not make it clear that the improvement meant here is the inclusion of Åland Sea into the configuration. We have rephrased this to be more clear.

---

## Author Comment (AC2)

**Author's response to RC2**

Westerlund, A., Miettunen, E., Tuomi, L., and Alenius, P.: Refined estimates of water transport through the Åland Sea, Baltic Sea, Ocean Sci. Discuss. [preprint], https://doi.org/10.5194/os-2021-56, in review, 2021

Below, reviewer comments are displayed with a gray background, while author responses are without highlighting.

**RC2: J. H. Reißmann, 02 Aug 2021**

> general comments:

> The authors configured a numerical model to investigate currents and resulting volume transports in the Åland Sea located between the Baltic proper and the Gulf of Bothnia in the northern Baltic Sea. This work is based on former model configurations with the aim to improve the performance of the the simulations in that challenging region with the focus on volume transports in unprecedented detail.

> Large parts of the manuscript are dedicated to model description and validation even outside the validation section in the results chapter. Also the results sections on currents and transports describe the simulations more or less in comparison with data and general knowlege in large parts and, consequently, do not contain much new facts. However, some new insight about seasonal current patterns, the surface transport and the pathways of deep water is given. Nevertheless, also the discussion focuses on the plausibility of the results, possible causes for deviations from existing findings, and ways to improve the model setup. Motivated by e.g. the importance of nutrient fluxes through the Åland Sea for the eutrophication status of the Gulf of Bothnia, unfortunately, neither implications of the findings on this nor the role of any relevant physical phenomena or driver for them and implications from that are discussed in detail and the manuscript remains more technical in this way. For this reason, the manuscript just has to be seen as a basework to develop modelling skills for the investigated region as the authors note themselves.

> The language is good and clear. The content has some repetions from the introduction throughout the intermediate chapters all the way to the conclusions. The figures are of good quality and easy to understand.

We appreciate the referee for providing this useful review and thank the referee. We are happy to note that the referee has found the language and the figures accessible and that the overall tone of the review is so positive.

These insightful comments from the reviewer made us realize that we had not articulated clearly enough the goals and scope of the manuscript in the introduction. We have made a number of revisions to the manuscript to clarify the objectives and to explain how this study

fits in the larger research plan we have for the coming years. We have also highlighted what new facts were expected.

Furthermore, to address these comments we extended the analysis of wind conditions. We have revised how modelled currents are analyzed. We have extended the discussion of transport rates, for instance. Also, a more comprehensive explanation of model validation was implemented.

We see this study as the first in a series of studies, which ultimately aim to answer the big questions related to water exchange in the target area. We agree that this work is basework in a sense that it is an important first step to address the big questions regarding water exchange in the study area. We do not feel this is a bad thing, but rather think that this is necessary for our efforts to provide in our future works the kind of in-depth analysis of water exchange in the target area that is currently missing.

We note that the main point of the paper, as set out in the title, is to provide refined estimates of transports through the Åland Sea, as we do. Previous efforts lack the kind of detail that is required for further study of this subject and we feel this study is a step that needs to be taken before further questions can be explored. We agree that the model validation is somewhat more comprehensive than would have been the bare minimum. We feel it is important to discuss the quality of the model results in detail here to build confidence in further studies we have planned.

We hope these changes, along with the ones detailed later, address the concerns raised by the referee. Thank you once again for these comments that help us improve this manuscript. We believe the manuscript has notably improved in this process. Please find our detailed answers to the specific comments made by the reviewer below.

 *specific comments:*

> *line 102: I think it would be good to briefly explain the physical meaning of the Samagorinsky parameter.*

We note that also referee #3 commented on how this parameter is presented here, and we agree that this section needed further polish. Based on their feedback, as well as this comment, we decided it is better to modify this section to remove the more technical information that is better available elsewhere, and to improve readability for the readers. Hopefully these changes address this concern.

> *line 107: Maybe some examples from literature should be given here.*

We have added some references.

> *line 163: Is there some literature proving this and the opposite for Föglö Degerby?*

We are not aware of any specific references for this, but have extended this paragraph to explain the situation better.

*> line 188: I agree that timing of the events seems good for which Figure 2 is appropriate. However, absolute differences are in the order of 0.1 m over most of the time shown. This would be much easier to see in a difference plot which is by definition much more appropriate to show differences. For me the following questions arise here: Is 0.1 m difference really quite small or what is to be called small here and why? And what is about the reference levels? Are they compareable at all? Are they determined in a comparable way? Can a target difference of 0 be expected from this point of view? Depending on your answers to these questions, it may be better to skip SSH differences as a mean for validtion, explain why a difference of 0 cannot be expected (mainly different reference levels I would assume) and focus validation on the timing and magnitude of the shown variations. Else, it should be explained why 0.1 m can be considered as small here and what implications these differences have on the dynamics and transports.*

Thank you for this comment. It made us understand that we had not explained fully how we had interpreted the SSH comparison, or what could be concluded from it. We have modified this subsection to explain which aspects of the SSH validation are the most relevant for the study at hand and to explain what kind of differences can be expected when comparing model vs. observation. Hopefully the subsection is better now.

*> line 198: Similar problem than before with SSH: What is the expected accuracy and why? Which accuracy is needed for the planned investigation to be reasonable?*

Thank you. Here too we understood that we had not explained the aim of this comparison in detail. We have modified this subsection to provide context on profile validation. Hopefully it has improved now.

*> line 206: Why (see comment on line 198))? Which implications does this have for the investigations?*

We do not believe this has significant implications for the time scales investigated in this paper, but thought it would be worthwhile to clearly indicate the limits of the applicability for the configuration. This enables the reader to evaluate the results and their applicability more realistically.

*> line 238 and before: How large is the error or accuracy of the numbers given in this paragraph and that one before?*

The challenge with this subsection was to find the right balance of accurate description of currents, and easy readability. After reviewing the paragraph in question and discussing this as well as other reviewer comments concerning this section, we came to the conclusion that this paragraph and Fig. 6 (now Fig. 5 in the new revision) were not the most easily accessible way of describing the essential information the reader needs to know about the accuracy of modelled currents. With this in mind, we revised Fig. 6 to present also the RMSE errors and modified this paragraph so that it more clearly describes the situation. Hopefully, taken together, this paragraph and Fig. 6 are now more accessible and relevant.

*> Figure 3: It is a bit a pitty that it is not possible for all profiles to unambiguously assign the corresponding NEMO profile to the monitoring profile. I thought about indicating*

*corresponding profiles from monitoring and model by colour, but for this the needed 12 to 32 different colours for all profiles is definitely too much and not manageable.*

We completely agree. In fact, we experimented with a number of different ways of presenting this data before initial submission, including color coding the profiles. In the end, we came to the conclusion that the current approach was the most usable for the reader.

> *Figure 5: General problem with these plots: They are not normalised to phase space. But here this seems not to be very relevant for the discussed findings.*

Thank you for this comment. If we understood correctly, no specific change was requested here.

> *line 266: A good measure to show that would be to present the magnitudes of the vector means normalised to the means of the vector magnitudes. The effect itself is trivial and simply geometric.*

We considered including persistency plots in the manuscript, which we assume the reviewer is referencing here. In the end, we decided not to include them and to explain this qualitatively instead, as the number of plots is already high. These plots are quite often incorrectly interpreted and we expect including them would lead to more misunderstandings. We agree that analysis of current persistency would be interesting and are planning it for future studies.

> *lines 341/343/348/355: The mesoscale is defined by the first baroclinic Rossby radius. It would be good to give a number of its absolute size in this region. I suspect it to be quite small in this region and the model showing only mesoscale structures. Is the model capable to simulate submesoscale structures reasonably at all?*

We are aware of relatively few sources that have investigated the baroclinic Rossby radius in the target area. Fennel et al. (1991) is probably the most commonly used reference. They estimated the Rossby radius for the neighbouring basins. For the Gulf of Finland their estimate was 1.3-2.5 km and for the Baltic Proper around 5 km. Alenius et al. (2003) gave values between 2-4 km in the GoF. Westerlund and Tuomi (2016) gave a rough estimate in the Bothnian Sea of the order of 3-5 km. The technical report by Dargahi and Cvetkovic (2014) gave estimates based on a model calculation with mean values as small as 1 km (SD 0.7 km) in the Åland Sill area, but much higher values (7.3 km, SD 1.5 km) in the Åland Sea proper and even in the Southern Quark (3 km, SD 1.1 km). We have also performed some rough calculations based on CTD data in order to understand better how these values given in the literature relate to reality. In general, we got a range of values that were of the same order but somewhat smaller than Dargahi and Cvetkovic (2014).

Based on the information available, there certainly remains some uncertainty and it is possible that in some coastal and sill areas the model is unable to fully resolve the submesoscale. But, in the main basins of the target area, we do not expect this to be the case. Given that we are not aware of any peer reviewed sources directly answering this question, an absolute value for the radius in the manuscript would likely oversimplify the

issue. Therefore we would prefer to leave it for future work. A closer investigation of this would certainly be well placed in a future work more closely discussing the submesoscale dynamics in the area.

References:
Alenius, P., Nekrasov, A., Myrberg, K., 2003. Variability of the baroclinic Rossby radius in the Gulf of Finland. Cont. Shelf Res. 23 (6), 563–573.
Dargahi, B., & Cvetkovic, V. (2014). Hydrodynamic and Transport Characterization of the Baltic Sea 2000-2009.
https://www.diva-portal.org/smash/get/diva2:897280/FULLTEXT01.pdf
Fennel, W., Seifert, T., & Kayser, B. (1991). Rossby radii and phase speeds in the Baltic Sea. *Continental Shelf Research*, *11*(1), 23-36.

*> line 457: This also raises the question how much bigger the model domain should be in relation to the investigated area. Would the model results be more robust against disturbances or biases in the boundary data if the model domain is somewhat larger than the investigated area? Why is the relation between both choosen as it is here?*

The selection of the model area always involves several factors such as computational requirements of the configuration, physical and topographical features of the area, user requirements, data availability for the chosen area, financial resources, and so on. This always involves some compromise. In this case, the presented configuration is intended to serve as a platform for future studies, which means that a wide array of requirements had to be taken into account. Furthermore, due to the iterative development cycle for these kinds of modelling configurations, any choice also includes some risks, as it is necessary to fix the modelling domain relatively early on in the development cycle and it is not possible to foresee all potential problems that could arise. Usually some problems do arise. Then it is necessary to weigh their impact against the impact, risks and potential cost of further changes to the modelling configuration. In this particular case we determined that the benefits of this kind of a change are relatively small when compared to the risks and costs. For this reason, the model domain was not further expanded at this time.

*> technical corrections:*

*> line 41: Witting (1908) is missing in the references.*

Fixed.

*> line 45:  Granquist (1938) is missing in the references.*

Fixed.

*> line 54: Is F64 correct? In Figure 1 station F69 is located in Lågskär Deep. If F64 is correct it should be emphasised somehow that a station quite far away was used to draw this conclusion. Suggestion: ,He also concluded from data from station F64, although it is located in the Åland Sea proper, that at Lågskär Deep …'*

The reviewer is right, this was a typo. Palosuo used data from F69, not F64. This has been fixed.

> *line 78: Suggestion: Change ‚big depth gradients' to ‚large depth gradients'.*

Fixed.

> *lines 206/407/434: Suggestion: Replace ‚sensible' by ‚reasonable'.*

Fixed.

> *line 268: Suggestion: ‚In the other winters' instead of ‚Of the other winters'*

This sentence no longer exists in the revised version.

> *line 334: Suggestion: Better write ‚are plausible' than ‚were plausible'. Check used times like this in the whole section! I would suggest to use present time to descripe your findings in general in the whole manuscript, especially also in section 3.*

Fixed. We have also checked verb tenses throughout the manuscript and modified them where necessary. We decided to mostly keep using past tense in the results section 3, as it is recommended by several scientific writing manuals. We do recognize that recommendations vary, and that the present tense is often a very good choice for describing the results.

> *line 352: Suggestion: Put ‚at our disposal' to the end of the sentence for better readability.*

Fixed.

> *line 361: Suggestion: ‚are' instead of ‚to be' to make it a correct sentence.*

Fixed.

> *line 372: Suggestion: ‚direction of the mean seasonal current' sounds more correct.*

Fixed.

> *line 386: Although the Knudsen theorem is commonly quite known, maybe is would be good to add the reference here.*

Done.

> *line 387: ‚note' instead of ‚notes'*

Fixed.

> *line 422: Suggestion: Ommit ‚than', just ‚less ideal' reads better and maybe is more*

*correct.*

Fixed.

*> line 426: Suggestion: ‚a minor issue' instead ‚less of an issue'.*

Fixed.

*> line 485: ‚flows' instead ‚flow'*

Fixed.

---

## Author Comment (AC3)

**Author's response to RC3**

Westerlund, A., Miettunen, E., Tuomi, L., and Alenius, P.: Refined estimates of water transport through the Åland Sea, Baltic Sea, Ocean Sci. Discuss. [preprint], https://doi.org/10.5194/os-2021-56, in review, 2021

Below, reviewer comments are displayed with a gray background, while author responses are without highlighting.

**RC3: Anonymous Referee #3, 07 Aug 2021**

> General comments:

> The authors have adapted a high-resolution model, which has already been used in previous studies, to the Åland Sea area in order to investigate in unprecedented detail the water transports between the Åland Sea and the Bothnian Sea in the north and the Baltic Proper in the south. Previous modelling studies as well as observational data lack detail, leaving open questions about, for example, the changing eutrophication status of the Bothnian Sea. This study now aims - at least to my reading - to fill these gaps in understanding (at least partially).

> After a clearly structured description of the methods and used data, the results chapter is unfortunately no longer clearly ordered, as the model validation is not limited to the model validation part, but comprises large parts of the rest of the chapter. Accordingly, this chapter unfortunately contains only a little new findings on water exchange in the Åland Sea. In the discussion section that follows, the focus is then on possible mainly model-related reasons for the deviation of the model results from measurements, again on validation against already known findings from older studies and on potential further improvements to the model setup. In conclusion, it does not seem so clear whether the focus of this study is on the water transport through the Åland Sea itself or on the demonstration that the model can reproduce this transport very well.

> All in all, the manuscript reads very well to me as a comprehensive model introduction or validation paper, describing a model that works in my opinion very well in the challenging region of the Åland Sea, and is certainly suitable for numerous and diverse follow-up studies in the region, as the authors themselves write in the outlook. In contrast, the analysis of water transport in the Åland Sea is unfortunately somewhat superficial. For example, no reasons for exceptionally occurring northward mean seasonal currents are discussed, and the knowledge gap on the change in the eutrophication status of the Bothnian Sea mentioned in the introduction is not even addressed in the discussion.

> I would therefore recommend the authors to carry out more in-depth and somewhat more comprehensive analyses on water transport on the basis of the already available data used for this study (possibly including the meteorological data) or to shift the focus/objective of this

*study more towards the applicability/quality of the chosen model setup.*

We would like to thank the referee for taking the time to review the manuscript and helping us improve it. These comments are highly appreciated. We are glad that the reviewer thinks the model works well, and are happy to hear that we were able to convey our understanding of its suitability for further studies.

After carefully considering the feedback given by this referee, we have implemented a number of modifications to the manuscript. We have clarified study focus and how it sits in the larger research plan we have for the coming years. Clearly, the explanation of objectives and scope of the manuscript needed polish. We have modified the introduction to articulate these more clearly.

Also, we added further analysis for meteorological conditions and the exceptionally occurring northward mean seasonal currents that the reviewer mentioned. Furthermore, we extended the analysis in the manuscript to include more discussion of e.g. transport rates. We have also revised how currents were analyzed. A more comprehensive explanation of model validation was included.

We see this study as the first in a series of studies, which ultimately aim to answer the big questions related to water exchange in the target area. We agree with the reviewer that there are still many questions that need answering that were not (fully) addressed in this paper. How these questions are split into different studies is of course somewhat subjective, and we certainly understand if the reviewer has a different point of view regarding this issue.

This particular paper lays groundwork for further studies we envision for the following years. We feel it is important to try to relieve concerns that are generally raised about modelling studies such as this one, and even more so in a region that is as challenging as the Åland Sea. Accordingly, some aspects of the manuscript seem, and in fact are, somewhat technical in nature, and the model validation is somewhat comprehensive. We note that the main point of the paper, as set out in the title, is to provide refined estimates of transports through the Åland Sea, and this is done.

We believe the manuscript has notably improved with the modifications we have implemented based on reviewer suggestions. Hopefully these changes, along with the changes made to answer the specific comment as detailed below, address the concerns raised by the reviewer.

*specific comments:*

> *line 21: Could the water transports through the Åland Sea explain the changes in eutrophication status of Bothnian Sea? If so, why? Are there changes in transports? It would be good to refer to this again in the discussion section.*

Thank you for this comment, which made us realize we clearly had not articulated clearly enough how this study relates to the important questions raised by the reviewer here. We expect to see an answer to these big questions in future years after further studies have been completed. We have modified the introduction to reflect this fact better.

*> line 103: Why is the Smagorinsky parameter "rn_csmc" the only parameter explicitly mentioned here? What makes the selection of this parameter (as opposed to other parameters) so important? This selection would have to be explained in much more detail. Alternatively, one could list all selected parameters as an appendix or similar and not mention it here at all.*

After consideration, we agree that the mention of this particular parameter is unnecessary and have condensed this paragraph accordingly.

*> line 109-114: I think thermodynamic formulation means an ice model without ice drift. What effect on water transports can be expected from ice drift (both in mild and in harsher winters) that it is explicitly stated that only mild winters occurred in the modelling period?*

The most relevant difference for the study at hand is that the thermodynamic module of the ice model handles momentum transfer from the atmosphere differently than the full dynamic ice model. Atmospheric stresses are transferred directly to the ocean, when in reality ice cover should make these stresses smaller. On the other hand, the atmospheric forcing originates from models that typically take sea ice concentration as input. In atmospheric models, ice cover may increase surface roughness and induce a stable boundary layer lowering wind speeds. We mention mildness of winters, because severe ice winters would introduce an additional source of uncertainty to the analysis, the effect of which might be challenging to quantify.

*> line 162/163: Considering the location of Turku and Forsmark, it is easy to imagine that these gauges are not representative for the main study area. However, this should be explained in more detail (e.g. also by plotting the position of the gauges in Fig. 1).*

We have extended this paragraph so that this is explained better. In this particular case we would prefer not to include these two locations in Fig. 1, as we do not actually use this data and the Figure already has many labels.

*> Fig. 2: The time series obviously shows a high correlation, but there also seems to be a relatively large bias, so that I would like to see the numbers of the statistical parameters in addition to the figures. Can this bias be explained by a bias in the boundary conditions only or are there other reasons?*

We noticed that we had not fully explained how the SSH validation had been conducted and which factors are most relevant for the study at hand. We have modified this subsection accordingly. Hopefully it now better explains the relevant factors and why we have not used bias to describe the data.

*> line 193 ff.: In contrast to the thermocline, the halocline is only mentioned in this subchapter in connection with a salinity bias above it. Since the halocline plays a very important role in the further course of the analysis, I would recommend describing the of the modelled halocline quality (location and gradient compared to the observations) in more detail. Due to the large number of individual profiles, this cannot be seen directly from Fig. 3 in my opinion.*

We have expanded this paragraph with details about the halocline in the individual profiles.

> *line 260f.: Why is it to be expected that the current velocities on the western side are stronger than on the eastern side. I would like some words of explanation.*

This fragment at the end of the sentence was likely a reference to earlier results in the literature concerning currents in the Åland Sea, which are not really relevant for this analysis. It seems that we have left this fragment in the submitted manuscript by mistake. We have therefore removed it. Thank you for pointing this out.

> *line 267-269: What is the reason for this special characteristic of the winter 2013/14 (and also of the winter 2016/17)? Were there exceptional wind conditions?*

Yes, the wind conditions differed. We modified the paragraph to reflect this.

> *line 326 f.: It would be good to explicitly mention here the issues that have been discovered in previous publications.*

We agree the wording in this paragraph was a bit unclear. We have rewritten the paragraph, hopefully it is more clear now what improvements were meant.

> *line 360: Why did not use a slightly larger model area if you suspect errors in your main study area due to conditions at the (relatively) near open model boundary?*

The selection of the model area always involves several factors such as computational requirements of the configuration, physical and topographical features of the area, user requirements, data availability for the chosen area, financial resources, and so on. This in turn always involves some compromise. In this case, the presented configuration is intended to serve as a platform for future studies, which means that a wide array of requirements had to be taken into account. Furthermore, due to the iterative development cycle for these kinds of modelling configurations, any choice also includes some risks, as it is necessary to decide the modelling domain relatively early on in the development cycle and it is not possible to foresee all potential problems that could arise. Usually some problems do arise. Then it is necessary to weigh their impact against the impact, risks and potentially significant cost of further changes to the modelling configuration. In this particular case we determined that the benefits of this kind of a change are relatively small when compared to the risks and costs. For this reason, the model domain was not further expanded at this time.

> *line 363 f.: I would have given the numbers of missing data in earlier chapters and not just in the discussion.*

We have now added this same information to the methods section.

> *line 371: Same as line 267-269: What is/could be the reason for these two "special" cases?*

Please see the answer regarding lines 267-269 above.

*> technical corrections::*

*> line 54: Is station F64 really meant here? According to Figure 1, station F69 (and not F64) is positioned in Lågskär deep, so in this context F69 is probably meant.*

Yes, the reviewer is absolutely correct. This has been fixed.

*> line 78: "big depth gradients" should be changed to "large depth gradients"*

Fixed.

*> line 206/407: I would prefer the use of "reasonable" instead of "sensible"*

Fixed.

*> line 234: Figure 6 is mentioned in the text before Figure 5 - I would recommend to change the numbering of these figures.*

Fixed.

*> line 238: The overestimation of the U-component should be between 0.006 and 0.034 (and not 0.34) m s-1.*

This number is no longer present in the revised version.

*> line 372: It must be "direction of the mean seasonal current"*

Fixed.

---

## Author Response (AR2)

**Author's response to reviewers, 2nd iteration**

Westerlund, A., Miettunen, E., Tuomi, L., and Alenius, P.: Refined estimates of water transport through the Åland Sea, Baltic Sea, Ocean Sci. Discuss. [preprint], https://doi.org/10.5194/os-2021-56, in review, 2021

Below, reviewer comments are displayed with a gray background, while author responses are without highlighting.

**RC1: Anonymous Referee #1, 19 Oct 2021**

> Review round #2 of submission os-2021-56

> General:
> In my first review, I suggested to take a more explanatory direction of the study by exploring (at least some of) the physical driving mechanisms for the simulated circulation features in the Aland Sea. I also gave plenty examples for interesting questions and possible causal relations to dive in. Unfortunately, none of these were incorporated by the authors. Instead, they decided to stay with a rather descriptive report of the simulated volume transports at different pathways and their comparison to previous estimates. This is a pity, as I believe in this way the study will mainly serve just as a reference for some transport rates in the Aland Sea area. Nevertheless, in the revised ms the authors have made this rather limited aim of the study more clear, compared to the first version. Hence, as a reader, I still think that the study could have been way more interesting, but I no longer feel disappointed due to unsatisfied expectations. Therefore, I can accept the authors decision.

We thank the reviewer for once again providing constructive feedback. We are happy that the reviewer feels the aim of the manuscript is more clear now and that the manuscript can be accepted subject to minor revisions. Hopefully we have been able to resolve the remaining issues in a satisfactory manner. Please find our detailed response below.

> Main:
> If I am not mistaken, there is an incorrect estimation of the flushing time of the Aland Sea basin presented in L338-340. The volume of the basin is divided by the net southward volume flux. Because of the opposite directions of the upper (southward) and deep (northward) circulation, the net southward volume flux could also be zero. Yet, the area would be permanently flushed. Thus, neglecting sink and source terms (e.g. local river runoff, precipitation, evaporation), the total lateral inflow or outflow of the basin has to be considered to properly calculate the flushing time.

Thank you for bringing up this issue. We introduced the calculation of flushing time in the previous iteration following a recommendation by a reviewer, but at that time we did not define terms or discuss the assumptions of this calculation. The addition was also left somewhat disconnected from the rest of the manuscript. Clearly, this was our mistake. As this calculation is of minor importance to the points discussed in the paper, and as proper interpretation of the flushing time and discussion of the underlying assumptions would

require a more complete and careful consideration, including definitions of what is actually being calculated, we decided that this calculation is best left to future manuscripts, where we plan to discuss other similar issues in depth. Thank you again for this useful comment and highlighting this issue.

> *Minor:*
> *I suggest to dedicate a separate section to the model evaluation, e.g. gathering sections 3.1 to 3.2.1 of the present ms version. Section 3.2.2 would the be the first result section.*

Thank you for this suggestion. Finding a way of making this change so that it would improve readability and not disrupt the flow of the text turned out to be difficult. It appears that implementing this change would make the structure of the results section hard to follow, as then the material about currents would be scattered across several subsections. Therefore we kept the original order of subsections in this version. We believe this is better for the overall readability of the manuscript.

> *Moreover, I strongly suggest to use present-tense when referring to the author´s own work and results and past-tense when referring to the work by others. This is an elegant way that allows the reader to easily identify which parts are new. Besides, it sounds odd when new research is presented as reporting on the past. A few examples are listed below.*

Thank you for the suggestion. We are not native English speakers, and apologize for any linguistic issues remaining in the manuscript, but note that there will be a proofreading stage after acceptance. As noted before, our guideline for the use of tenses is advice from scientific writing manuals. e.g. nature.com (which quotes Doumont (2010)), Day (1998) or Schultz (2009).

According to our understanding, the reviewer suggests that it would be better to use present tense also for our own work. This advice differs from that given in the references. However, we see that the present tense is often a very good choice for describing the results. We are happy to reconsider if presented with differing authoritative views e.g. by the proofreaders.

References:
Day, R. A. (1998). How to write and publish scientific papers.
Doumont, J., ed. English Communication for Scientists. Cambridge, MA: NPG Education, 2010. See https://www.nature.com/scitable/topicpage/effective-writing-13815989/
Schultz, D 2009, Eloquent Science : A Practical Guide to Becoming a Better Writer, Speaker, and Atmospheric Scientist, American Meteorological Society, Boston, MA.

> *L11-12: Use present-tense.*

Please see our answer regarding tenses above.

> *L31: "topographic gradients"*

Fixed.

> *L33-34: Use 2x "has a maximum".*

Fixed.

> *L43: "to investigate exchange fluxes through this area."*

Fixed.

> *L84: "in other regional configurations"*

Our understanding is that (at least in Baltic modelling community) the term "regional" most often refers to configurations with a domain covering a larger area than one sub-basin. For this reason we did not add the word "other".

> *L92: Suggest to move L92-95 to L75. The technical aspects explained in L75-91 would become a clearer context.*

Done, thank you for the suggestion.

> *L95: Delete "eventually".*

Fixed.

> *L104: Use present-tense: "The simulated time span covers ..." (also L106)*

Please see our answer regarding tenses above.

> *L116: "sea ice model"*

Fixed.

> *L123: In the first integral, shouldn´t it be only a single integral symbol, as you are integrating over dA?*

We apologize if it was not clear from the context, but this integral uses the double integral notation, for which two integral symbols are commonly used.

> *L133: "has a resolution"*

Fixed.

> *L134-148: I still feel that this is largely redundant information for the reader and should be condensed to e.g.: "After interpolation of the bathymetric source data, the resulting numerical grid was smoothed with a Gaussian filter with standard deviation of 1.2 grid points to weaken the steepest bathymetry gradients and ensure numerical stability."*

We would argue that this information is not redundant but one of the more important methodological features of this study. It is commonplace in Baltic Sea modelling studies to see model bathymetries that are not checked by hand at all, but rather just algorithmically

processed. This frequently results in issues in model results that could have been avoided, and limits the usability of model results for further studies. These issues often manifest themselves especially in coastal and shallow areas, of which there are plenty in our modelling domain. We have ourselves encountered several modelling datasets that are not usable in certain coastal areas because of such issues. The suggested reformulation would leave the reader with the impression that we had used only algorithmic processing, from which the reader could incorrectly infer that model data is of lower quality than in reality. We have reviewed and edited these paragraphs again to remove redundant information, but strongly feel that most of the information is valuable for readers tackling similar issues.

> L158: Does this mean that the barotropic velocities prescribed at the open lateral boundaries do not resolve semi-diurnal tidal currents?

We expect that with current boundary conditions, the model can't be expected to resolve periodic processes on timescales of hours.

> L207: Not sure whether "clines" is a valid expression. I suppose you mean thermocline and halocline.

We modified the sentence to be more clear and explicitly state what was meant.

> L210: "generally" instead of "typically"

Fixed.

> L210: "This includes the strength of the thermal stratification and its vertical position, ..."

Fixed. We also made a similar modification at L207, where a similar phrase was used.

> L235: "permanent halocline"

Fixed.

> L236: "an intermediate layer between the thermocline and halocline"

Fixed.

> L237: What is meant by "more pronounced" here? You mean "thicker"?

We clarified this sentence.

> L251: "(representing the deepest model layer at the ADCP station)"

Fixed.

> L256: "The largest bias and RMSE in the current magnitude occur at depths of the halocline, with up to ..."

Fixed.

*> L260: southern*

Fixed.

*> L262-263: 2x "northward flowing currents"*
*In fluid dynamics, the term "northerly flow" usually refers to a southward direction of the flow*
*(e.g. we use to say westerly winds when the winds are blowing from west to east).*

Fixed.

*> L272: Regarding the model biases: If my understanding of the model boundary conditions*
*is correct (L158), the prescribed daily barotropic velocities could be an issue here as these*
*are not resolving semi-diurnal tidal currents, in contrast to the used hourly SSH.*

We do not expect issues with tidal currents to be significant in this area. See e.g. Medvedev et al. (2013) for information regarding tides in the investigation area.

Reference:
Medvedev, I.P., Rabinovich, A.B. & Kulikov, E.A. Tidal oscillations in the Baltic Sea. *Oceanology* **53,** 526–538 (2013). https://doi.org/10.1134/S0001437013050123

*> L280: "most frequent"*

Fixed.

*> L286: Please make more clear that here you are not referring to the cyclonic recirculation*
*mentioned in the previous 2 sentences (where currents are stronger at the eastern side,*
*L302).*

We clarified this sentence.

*> L331-332: "... as precipitation and evaporation roughly balance to net zero freshwater flux*
*at the sea surface." (true?)*

We clarified this sentence.

*> L340: As mentioned above, the flushing time of 6.5 months is incorrect in my opinion. As*
*the Aland Sea is a 2-layer system, both the upper and lower circulation contribute to water*
*exchange. Therefore, one cannot use the net flow of 24000 m3 s-1 to calculate the flushing*
*time.*

Thank you, please see our answer in the beginning of this letter.

*> L373ff: Another example for the odd use of tenses. Please use present-tense.*

Please see our answer regarding tenses above.

> *L400 and 403: Suggest 2x "possible" instead of "plausible".*

Fixed.

> *L426: "It is challenging ..."*

Fixed.

> *L450: "investigation" instead of "examination"*

Fixed.

> *L451: "simulates/captures" instead of "interpreted"*

Fixed.

> *L467: "relevant" instead of "bothersome"*

Fixed.

> *L470: Well, the surface layer thickness of standard z coordinates is mainly determined by SSH variations due to tides, which is not relevant in the Baltic. Sea ice thickness is another issue, though you did not have extreme winters in your study period.*

Thank you for this comment. If we understood correctly, no specific change was requested here.

> *L477: By "steeper gradients" you mean strong vertical gradients in the water column? Please clarify.*

This sentence has been clarified, thank you.

---

## Author Response (AR3)

**Author's response, 3rd iteration**

Westerlund, A., Miettunen, E., Tuomi, L., and Alenius, P.: Refined estimates of water transport through the Åland Sea, Baltic Sea, Ocean Sci. Discuss. [preprint], https://doi.org/10.5194/os-2021-56, in review, 2021

Below, reviewer comments are displayed with a gray background, while author responses are without highlighting.

**Topic Editor comments, 12 Nov 2021**

> **Comments to the author**:
> Dear Authors
> I am pleased to accept your manuscript for publication subject to technical corrections. In general, the language is very good but please check the usage of "the, a" which is difficult (I admit) if the native language differs in that respect. In addition, it should be "1950s" instead of "1950's" and "Baltic proper" instead of "Baltic Proper" (no proper name).
> Thank you and best wishes
> Markus Meier

Dear Professor Meier,

Thank you for your comments. We have fixed the specific language issues you mentioned and also made our best effort to correct the issues with definite and indefinite articles. Hopefully the English language has improved now.

Best regards,
the authors